JCB Journal of Cell Biology

# RalGTPases contribute to Schwann cell repair after nerve injury via regulation of process formation

Jorge Galino[1]*, Ilaria Cervellini[1]*, Ning Zhu[1], Nina Stöberl[1], Meike Hütte[1], Florence R. Fricker[1], Garrett Lee[1], Lucy McDermott[1], Giovanna Lalli[2], and David L.H. Bennett[1]

**RalA and RalB are small GTPases that are involved in cell migration and membrane dynamics. We used transgenic mice in which one or both GTPases were genetically ablated to investigate the role of RalGTPases in the Schwann cell (SC) response to nerve injury and repair. RalGTPases were dispensable for SC function in the naive uninjured state. Ablation of both RalA and RalB (but not individually) in SCs resulted in impaired axon remyelination and target reinnervation following nerve injury, which resulted in slowed recovery of motor function. Ral GTPases were localized to the leading lamellipodia in SCs and were required for the formation and extension of both axial and radial processes of SCs. These effects were dependent on interaction with the exocyst complex and impacted on the rate of SC migration and myelination. Our results show that RalGTPases are required for efficient nerve repair by regulating SC process formation, migration, and myelination, therefore uncovering a novel role for these GTPases.**

## Introduction

Peripheral nerve injury results in marked transcriptional and phenotypic changes within Schwann cells (SCs), which are critical for effective nerve repair (Jessen et al., 2015; Jessen and Mirsky, 2016). SCs are involved in every phase of the initial nerve injury response and required for subsequent repair (Chen et al., 2007). After injury, the distal stump undergoes Wallerian degeneration (Griffin et al., 2013). Axonal death triggers SC dedifferentiation (Arthur-Farraj et al., 2017; Clements et al., 2017). SCs contribute to the clearance of axonal- and myelin-derived debris (Gomez-Sanchez et al., 2015). SCs proliferate and fill the empty endoneurial tubes in organized longitudinal columns called bands of Büngner. These "repair" SCs undergo significant morphological transitions as they elongate and guide axons back to their targets (Fazal et al., 2017). There is a close association between SC processes and axons (Arthur-Farraj et al., 2012). Following complete nerve transection, SCs migrate and form bridges to aid the crossing of axons from proximal to distal nerve stumps (Parrinello et al., 2010). Finally, SCs envelop the regenerated axons, large-diameter axons are remyelinated, and small-diameter axons will be enclosed in SC "pockets" of nonmyelinating (Remak) SCs (Arthur-Farraj et al., 2012). This supportive environment is critical for successful axonal regeneration (Grinsell and Keating, 2014), and these events require extensive changes in SC process formation and extension.

Ral small GTPases, encoded by the *Rala* and *Ralb* genes, are members of the RAS superfamily of small GTPases (Bodemann and White, 2008). Like all GTPases, Ral proteins transduce signals in cells by cycling between an active GTP-bound and an inactive GDP-bound state (Feig, 2003). Overall, the RalGTPases RalA and RalB share 85% of sequence identity (Neel et al., 2011). RalA and RalB can have redundant functions (Peschard et al., 2012). The role of RalGTPases in the cells is due to their distinct subcellular membrane localization: RalA is found in the plasma membrane, cytoplasm, and perinuclear area, usually within endosomes, while RalB is primarily endosome-associated (Shipitsin and Feig, 2004). Also, their localization can be regulated by the activation state and phosphorylation (Cascone et al., 2008; Neel et al., 2011). RalGTPases have been implicated in a variety of neuronal processes, such as neural tube closure, neurite branching, cytoskeletal reorganization, and membrane dynamics (Lalli and Hall, 2005; Peschard et al., 2012; Das et al., 2014). The role of RalGTPases in glia has not been examined; however, they can interact with a number of downstream pathways, which could have an important impact on SC function. RalGTPases' effector Ral binding protein 1 (RalBP1) can deactivate cdc42 and Rac1 in the plasma membrane via intrinsic GTPase-activating activity, dephosphorylating cdc42 and Rac1 to the inactive GDP state (Matsubara et al., 1997). These small GTPases are both important for

[1]The Nuffield Department of Clinical Neurosciences, University of Oxford, John Radcliffe Hospital, Oxford, UK; [2]Wolfson Centre for Age-Related Diseases, King's College London, Guy's Campus, London, UK.

*J. Galino and I. Cervellini contributed equally to this paper; Correspondence to David L.H. Bennett: david.bennett@ndcn.ox.ac.uk.

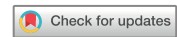

developmental myelination and remyelination after nerve injury in adulthood (Guo et al., 2012, 2013).

A further important pathway through which Ral GTPases can impact on membrane trafficking and cell migration is via the exocyst complex, an octameric protein complex involved in the tethering and spatial targeting of vesicles to the plasma membrane before vesicle fusion (TerBush et al., 1996; Sugihara et al., 2002; Liu and Guo, 2012). This complex has been shown to have a role in polarized delivery of cargoes in a number of cell types, including epithelial cells and neurons (Vega and Hsu, 2001; Yeaman et al., 2001; Lalli and Hall, 2005; Feng et al., 2012; Das et al., 2014). Active RalGTPases regulate the exocyst complex by direct binding to two of its components: Exoc2 and Exoc8 (Moskalenko et al., 2002). We have investigated the role of RalGTPases in SCs during nerve injury and repair. We have found that RalGTPase signaling is important for multiple aspects of regeneration, including remyelination and muscle reinnervation. RalGTPase signaling has an important role in the formation and extension of SC processes, SC migration, and axon myelination, which occurs in an exocyst-dependent manner.

## Results

### RalGTPases expression and activation state after nerve injury

We examined RalGTPases expression in the regenerating distal nerve stump following sciatic nerve crush (Fig. 1). We analyzed three time points after nerve injury: (1) 4 d, when SCs dedifferentiate and proliferate, macrophages infiltrate the nerve, and myelin clearance occurs; (2) 11 d, when SCs organize in the bands of Bungner and remyelination of axons by SCs begins; and finally, (3) 28 d, when almost all the axons are reinnervating their targets and are remyelinated (Fig. 1 A). There was a significant decrease in the levels of RalA at 11 d and a nonsignificant trend to decrease at 4 and 28 d after crush compared with noninjured nerves (Fig. 1, B and C). In contrast, the levels of RalB were significantly increased at all the time points analyzed following injury versus control (Fig. 1, B and E). As has been described in the literature, changes in the expression are not necessarily correlated with activation status of RalGTPases (Peschard et al., 2012); therefore, we measured the amount of activated GTP-RalA plus RalB after injury. In contrast to total expression levels of RalA, there was a trend to increased RalA activation across all time points, although this did not reach significance (Fig. 1, B and D). RalB activation state after injury showed a trend to decrease at 4 d and a significant decrease at 28 d (Fig. 1, B and F). Thus, expression and activation state of RalGTPases are not correlated after nerve injury, so we investigated the effect of genetic deletion of RalGTPases in the nerve injury paradigm to gain further insight into their role in nerve regeneration.

Given that SCs are a major cellular component of the peripheral nerve and the significant membrane remodeling required for remyelination, we examined the expression of RalGTPases in this cell type, staining with RalA antibody in cultured primary rat SCs. The IHC showed RalA in the perinuclear cytoplasmic region and cell nucleus, as described in other cell types. Notably, however, there was also enrichment of staining in the membrane edge of the lamellipodia (Fig. 1, G and G').

### Global ablation of RalB or SC-specific ablation of RalA are dispensable for axon regeneration, remyelination, and functional recovery after peripheral nerve injury

To study the role of RalA and B in peripheral nerve regeneration, we initially examined these molecules individually after crush (Fig. S1 A and Fig. S2 B). $RalB^{-/-}$ mice have a global ablation of RalB protein (Peschard et al., 2012), which we observed in the sciatic nerve (Fig. S1 B). Ultrastructural analysis of the distal sciatic nerve 1 mo after crush demonstrated no significant difference between $RalB^{-/-}$ and WT littermate controls in the number of myelinated or unmyelinated axons (Fig. S1, C–F), myelin morphology (Fig. S1, G and I), axon diameter (Fig. S1 H), or Remak bundle structure (Fig. S1, J and K). There was no significant change in internodal length (Fig. S1 L) or reinnervation of the neuromuscular junction (Fig. S1 M). Consistent with the anatomical findings, there was no significant difference in the rate of functional recovery assessed using the sciatic functional index (SFI), a measure of motor function (Fig. S1 N), and the pin prick test which is a reflection of sensory afferent regeneration to the skin (Fig. S1 E).

Global deletion of RalA is embryonically lethal due to exencephaly (Peschard et al., 2012). We therefore used a conditional approach focusing on the role of RalA in SCs. We crossed $RalA^{f/f}$ mice with $PLPCreER^{T2}$ to get $PLPCreER^{T2};RalA^{fl/fl}$ (herein called cRalA). In these mice, tamoxifen-inducible Cre expression is driven by the PLP (proteolipid protein) promoter, which is expressed in SCs. Recombination of the conditional RalA alleles after tamoxifen induction in cRalA led to a strong reduction in RalA protein expression levels in sciatic nerve lysates from mutant mice in both noninjured and injured nerves (Fig. S2 A). The residual expression of RalA was likely due to the expression of this protein in fibroblasts and axons, as well as a few nonrecombined SCs. We also studied for possible compensation in RalB, analyzing the expression of RalB in cRalA mice by Western blot in noninjured and injured nerves (Fig. S2 A). The expression of RalB did not change in either ipsilateral or contralateral nerves (Fig. S2 A). There was no difference between cRalA and control mice in nerve repair in any of the anatomical or functional outcomes measured (Fig. S2, C–O). In summary, RalA (conditionally ablated in SCs) and RalB (global deletion) are dispensable for nerve repair when studying the mutant alleles individually.

### Efficient ablation of RalA on $cRalA/B^{-/-}$ mice

To study the effect of RalGTPases on nerve repair, we used a combination of the knockout (KO) for RalB ($RalB^{-/-}$) with the SC conditional KO of RalA (cRalA), here named $cRalA/B^{-/-}$ mice. Since cRalA is a conditional model, we investigated RalA ablation in the $cRalA/B^{-/-}$ mice (Fig. 2 A). In both noninjured and injured mice, there was a significant reduction in RalA protein as assessed using Western blot (Fig. 2 B). In addition, immunostaining of teased fibers was performed to assess localization of RalA. As shown in Fig. 2 C in control mice, RalA was predominantly localized in SCs (in both noninjured and injured nerves). We

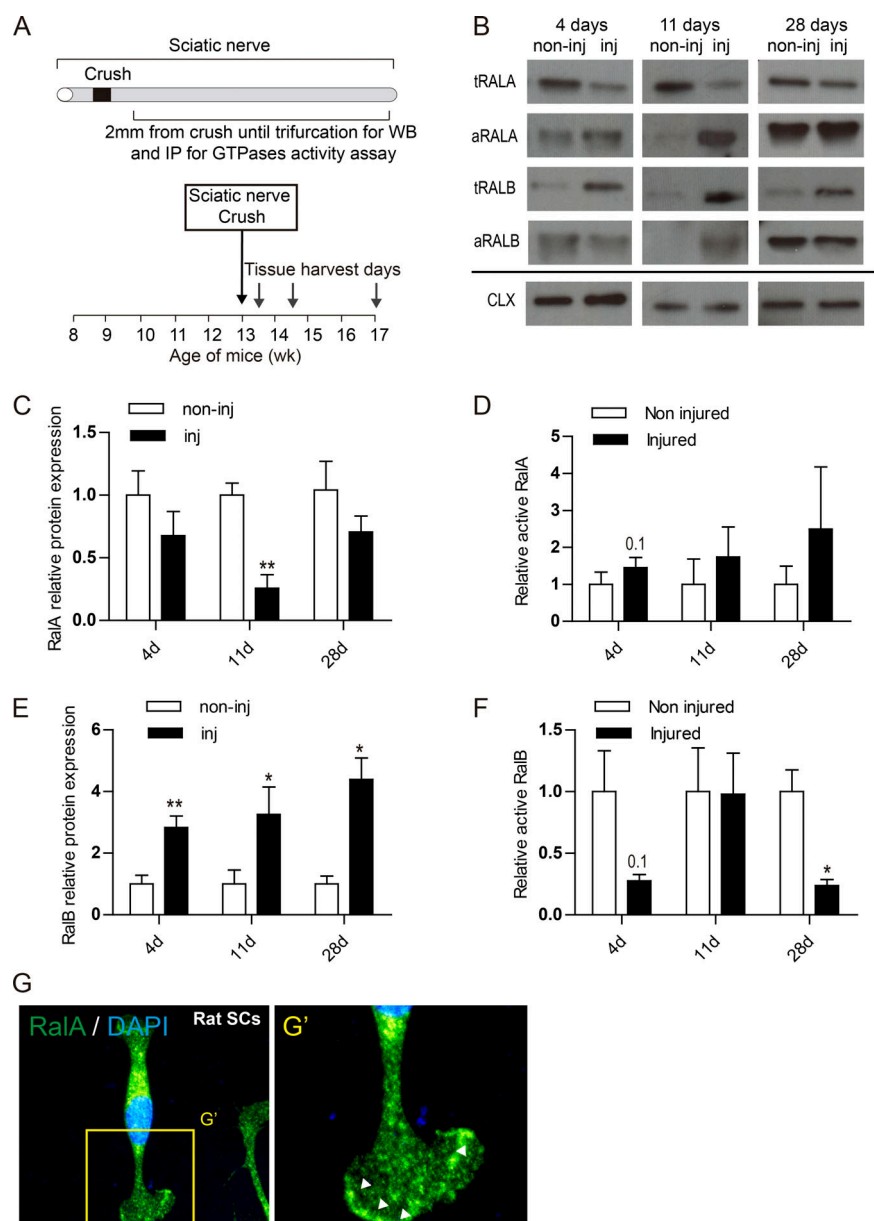

Figure 1. **Dysregulation of RalGTPase expression, activation after nerve injury and their expression in SCs in vitro. (A)** Schematic representation of the sciatic nerve crush model used to assess the role of RalGTPases on nerve repair. **(B)** Western blot of injured (inj) and noninjured (non-inj) sciatic nerves 4, 11, and 28 d after sciatic nerve crush in WT mice probed with antibodies against RalA and RalB. Calnexin (CAL) used as loading control. Immunoprecipitation blot of activated forms of RalA and RalB. **(C–F)** Quantification of the relative protein expression of RalA (C) and RalB (E) across the time points analyzed in the Western blots. Quantification of the relative protein expression of active RalA (D) and active RalB (F) across the time points. Data are presented as fold change against noninjured relative expression (**, P < 0.01; *, P < 0.05; Student's t test), n = 4 or 5. Data are presented as mean ± SEM. **(G)** Primary cultures of rat SCs demonstrate expression of endogenous RalA (green) in the nucleus, perinuclear cytoplasm, and in the plasma membrane. **(G′)** High magnification picture shows an enrichment of RalA in the membrane edge of lamellipodia (arrow bars). Scale bars, 50 µm.

also confirmed the ablation of RalA in SCs in the *cRalA/B⁻/⁻* mice (Fig. 2 C).

## SC-specific ablation of RalA in *RalB⁻/⁻* mice results in a hypomyelination of axons, reduced target reinnervation, and a delay in the functional recovery

To focus on SC function after nerve injury, we generated *cRalA/B⁻/⁻* mice in which the expression of RalA in SCs is conditionally ablated in *RalB⁻/⁻* mice. One month after tamoxifen treatment, *cRalA/B⁻/⁻* were completely healthy. 2 mo after tamoxifen dosing and 1 mo after crush, the sciatic nerve contralateral to the injury was used to assess the impact of RalGTPases on uninjured nerve. Ultrastructural analysis revealed no difference in myelinated and nonmyelinated axon count or g-ratio (Table 1) between tamoxifen-treated *RalA^fl/fl;RalB⁻/⁻* Cre-negative (hereafter referred to as "ctrl") and *cRalA/B⁻/⁻* mice.

12 d after nerve crush, many axons 2 mm distal to the crush site were observed undergoing remyelination in ctrl mice. In contrast, in *cRalA/B⁻/⁻* mice, we noted more axons with a diameter of >1 µm that were unmyelinated (Fig. 3 A and Table 2). Furthermore, the axons that were remyelinated had thinner myelin sheaths reflected in a considerable and significant increased g-ratio (Table 2). At the later time point of 1 mo after injury, the g-ratio and the number of unmyelinated axons with a diameter of >1 µm was still higher in *cRalA/B⁻/⁻* than in the ctrl mice (Fig. 3 A and Table 2). The number of axons in the distal sciatic nerve (Fig. 3, B and C), the C-fibers per Remak bundle, and the number of Remak bundles in the injured nerve did not significantly differ between ctrl and *cRalA/B⁻/⁻* (Fig. 3 and Table 2) at both time points analyzed. There was a significant decrease in the internodal distance in teased myelinated nerve fibers 1 mo after injury in *cRalA/B⁻/⁻* mice versus ctrl littermates

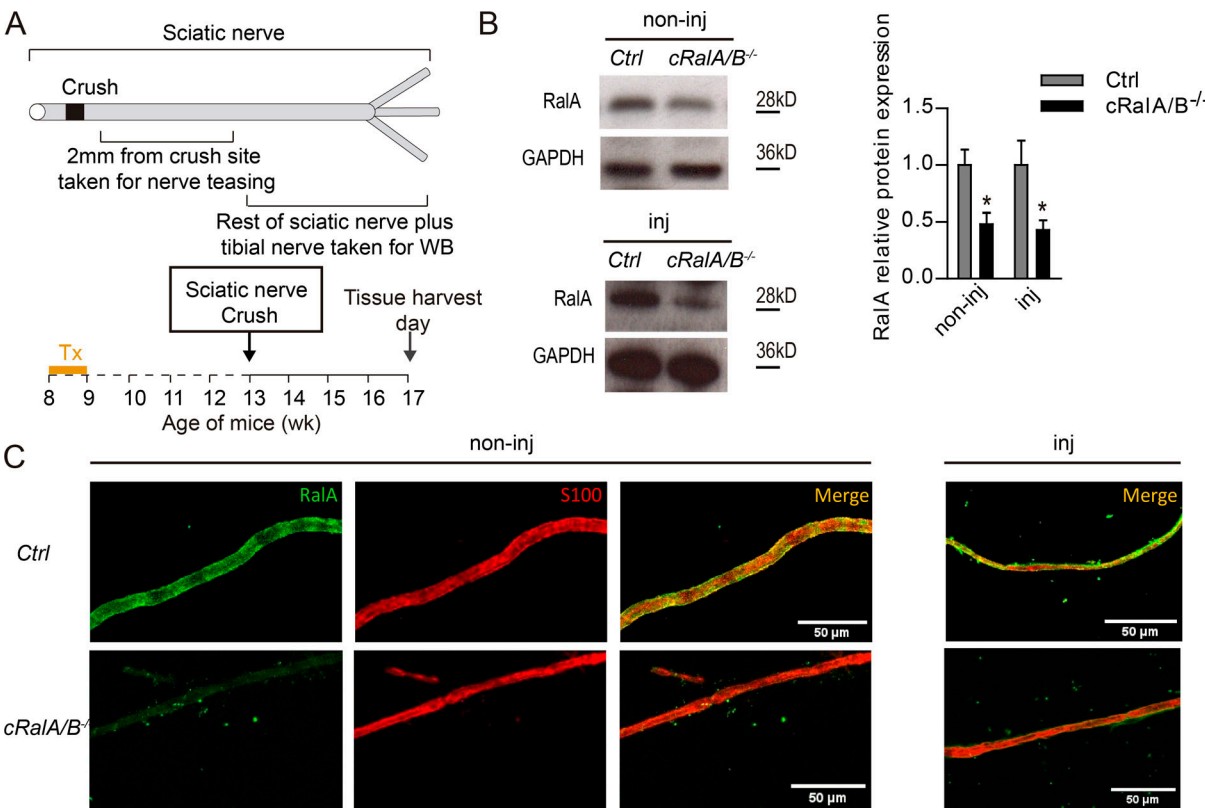

Figure 2.   **Analysis of RalA expression in *cRalA/B^−/−* mice. (A)** Schematic representation of the nerve crush experiment. Tx, Tamoxifen. **(B)** Western blots of noninjured and injured nerves showing RalA expression in ctrl and *CRalA/B^−/−* mice. GAPDH used as loading control. Data are presented as mean ± SEM. **(C)** Teased fibers staining of noninjured (non-inj) and injured (inj) nerves 4 wk after injury, ctrl versus *cRalA/B^−/−*. RalA is labeled in green, and S100 for SCs localization is labeled in red. RalA is predominately expressed in SCs, and it is ablated after tamoxifen treatment. *n* = 3.

(Fig. 3, B and C). To summarize, in the absence of RalA and B in SCs, there is reduced axon remyelination characterized by more large axons that are unmyelinated, increased g-ratio, and shorter internodes. To test whether disrupted RalGTPase expression led to a change in the target reinnervation by motor axons, we assessed innervation of neuromuscular junctions within the gastrocnemius muscle 4 wk after injury. We noted impaired reinnervation of the neuromuscular junction in *cRalA/B^−/−* mice versus ctrl as determined by a reduced occupancy of the postsynaptic acetylcholine receptors (revealed by α-Bungarotoxin labeling) by presynaptic motoneuron terminals (Fig. 3, D and E). Although we did not observe changes in the number of axons in the nerve trunk, *cRalA/B^−/−* mice do show impaired reinnervation of distal targets.

In assessing functional recovery following injury, in both *cRalA/B^−/−* and ctrl mice, there was a marked drop in SFI immediately after crush that reached the same nadir in both genotypes. *cRalA/B^−/−* mice had a slower recovery of the SFI following injury versus ctrl (which significantly differed from ctrl at days 12 and 14; Fig. 3 F). The pin prick test was comparable between ctrl and *cRalA/B^−/−* littermates (Fig. 3 G).

## Macrophage recruitment, myelin clearance, and SC proliferation are not altered following nerve injury in *cRalA/B^−/−* mice

Early events in Wallerian degeneration following peripheral nerve injury are the de-differentiation of the SCs, recruitment of macrophages, and subsequent removal of myelin debris by both cell types. To examine whether RalGTPases expression in SCs plays a role in these processes, macrophages and myelin ovoids contained within macrophages and SCs were counted in both tamoxifen-treated *cRalA/B^−/−* and ctrl mice (Fig. 4, B–D) 4 d after crush. As shown by micrographs analysis, in both mutant nerves, there were clear features of Wallerian degeneration with macrophage infiltration and collapsed myelin resulting in ovoid formation (Fig. 4 B). However, there was no change in the number of macrophages (Fig. 4 C), nor in the number of myelin ovoids contained within them (Fig. 4 D) between genotypes.

Table 1.   **Summary of nerve anatomy measurements in non-injured nerves**

| Non-injured | Ctrl | cRalA/B^−/− |
|---|---|---|
| Myelinated axons × 1,000/mm² | 7.15 ± 0.72 | 7.65 ± 0.26 |
| Unmyelinated axons × 1,000/mm² | 46.78 ± 8.66 | 48.68 ± 4.18 |
| g-ratio | 0.845 ± 0.005 | 0.879 ± 0.006 |

Quantification of the total number of myelinated axons, unmyelinated axons, and g-ratio was unchanged in non-injured nerves of *cRalA/B^−/−* versus ctrl. Number of animals = 4. Number of quantified fibers per animal at 12 d = 50. Number of quantified fibers per animal at 1 mo = 150.

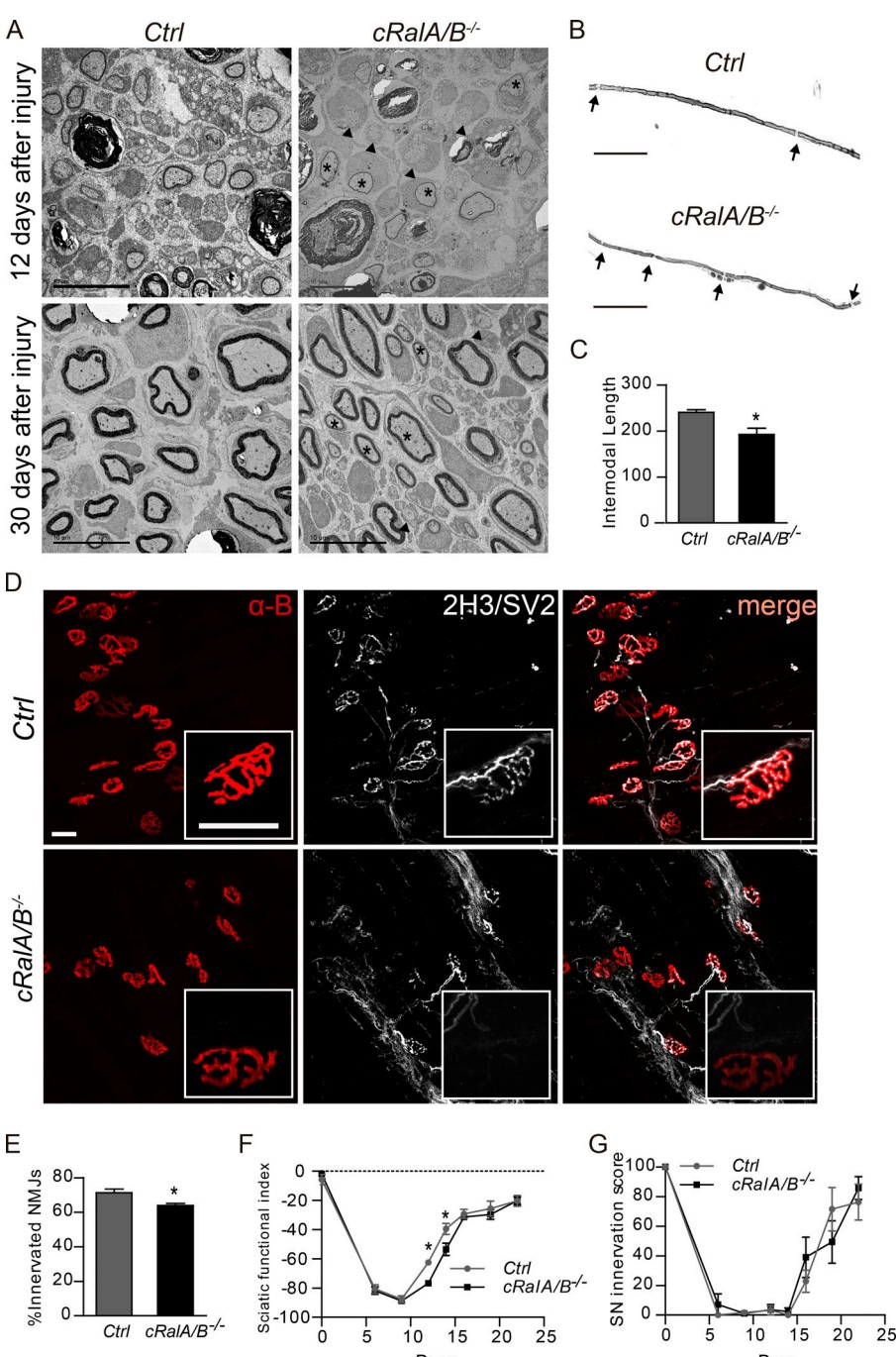

Figure 3. **RalGTPases are required for re-myelination, neuromuscular junction rein-nervation, and optimum functional recovery following peripheral nerve injury. (A)** Electron micrographs of transverse sections of the distal sciatic nerve 12 d and 1 mo after crush in control (Ctrl) and cRalA/B[−/−]. At 12 d in ctrl mice, most axons are undergoing remyelination, whereas in cRalA/B[−/−] mice, some axons with >1 μm diameter are still without myelin (arrowheads), and the ones that are myelinated have thinner myelin sheaths (black asterisk). **(B)** Representative photomicrographs of teased nerve fibers stained with osmium to analyze internodal distance of the distal sciatic nerve 1 mo after crush in ctrl and cRalA/B[−/−] mice. **(C)** Quantification of in-ternodal distance, *, P < 0.05, Student's t test, n = 4. **(D)** Representative photomicrographs of neuromuscular junctions within the gastrocne-mius muscle of cRalA/B[−/−] and ctrl. SV2 and 2H3 label presynaptic motoneuron axons and termi-nals (white), and α-bungarotoxin (α-B) labels post-synaptic acetylcholine receptor (AchR; red). Highlighted in the white squares are represen-tative high-magnification pictures of neuromus-cular junction endplates stained for SV2/2H3 and α-bungarotoxin; there is a lack of reinnervation in the cRalA/B[−/−] endplate. **(E)** Quantification of the number of neuromuscular junctions (NMJ) reinnervated 1 mo after nerve injury. *, P < 0.05, Student's t test, n = 5. **(F)** cRalA/B[−/−] mice pre-sent a significant delay in the functional recovery between 12 and 14 d after injury assessed using the SFI compared with control littermates (*, P < 0.05, Student's t test, n = 8–12). **(G)** Assessment of response to pin prick stimulation as a measure of reinnervation of the paw by nociceptive afferents. n = 8–12. Data are presented as mean ± SEM. Scale bars, 10 μm for electronic micrographs, 100 μm for teased fibers, and 50 μm for neuromuscular junctions.

Also, there was no change in the myelin basic protein (MBP) immunoblot between genotypes (Fig. 4, E and F), indicating that there is not a delay in myelin clearance and macrophage re-cruitment following peripheral injury in cRalA/B[−/−] mice com-pared with ctrl mice.

We analyzed the expression of the proliferation marker Ki67 in sciatic nerve longitudinal sections of cRalA/B[−/−] and ctrl mice 12 d after injury. We did not detect any difference in cell pro-liferation between cRalA/B[−/−] and ctrl mice (Fig. 5, A and B). We also performed a proliferation assay in vitro in cultured SCs from WT and cRalA/B[−/−] mice. Cultured mice SCs from both genotypes were treated with 4-hydroxytamoxifen (4-OHT; a

tamoxifen metabolite used for in vitro experiments) for 3 d. 2 d later, we stopped the treatment to see if this was effective in reducing RalA and B expression (Fig. 5 C). Tamoxifen-treated cRalA/B[−/−] showed a significant decrease in RalA expression compared with WT mice assessed by immunoblot for RalA. We also noted that RalB protein expression in cRalA/B[−/−]-derived cultured SCs is decreased compared with WT (Fig. 5, D–F). We did not observe any significant change in proliferation, assessed by immunostaining for the proliferation marker Ki67, in cRalA/B[−/−] compared with WT (Fig. 5, G and H). These results suggest that RalGTPases are not implicated in SC proliferation in vitro or after nerve injury.

**Table 2. Summary of morphometry in injured nerves**

| Injured | 12 d after injury | | 28 d after injury | |
|---|---|---|---|---|
| | Ctrl | cRalA/B$^{-/-}$ | Ctrl | cRalA/B$^{-/-}$ |
| Myelinated axons × 1,000/mm$^2$ | 7.15 ± 0.72 | 7.65 ± 0.26 | 19.10 ± 1.70 | 16.28 ± 1.60 |
| Unmyelinated axons × 1,000/mm$^2$ | 46.78 ± 8.66 | 48.68 ± 4.18 | 50.33 ± 12.81 | 30.15 ± 6.73 |
| % nude axons >1 µm | 12.55 ± 1.21 | 25.71 ± 2.30* | 4.57 ± 0.64 | 9.56 ± 1.18* |
| g-ratio | 0.845 ± 0.005 | 0.879 ± 0.006* | 0.657 ± 0.009 | 0.693 ± 0.008* |
| Remak bundles × 1,000/mm$^2$ | 6.25 ± 1.84 | 5.35 ± 0.81 | 6.63 ± 0.87 | 6.85 ± 0.30 |
| % bundles with polyaxonal pockets | 39.45 ± 2.91 | 50.98 ± 3.83 | 14.75 ± 3.55 | 15.92 ± 2.47 |
| C-fibers/Remak bundle | 8.39 ± 0.85 | 7.91 ± 0.42 | 5.8 ± 0.30 | 6.12 ± 0.60 |

Quantification of the total number of myelinated and non-myelinated axons per sciatic nerve cross-section at 12 and 28 d after crush does not show any difference between mice. Quantification of non-myelinated axons of >1 µm diameter at 12 and 30 d after injury shows a clear increase in the percentage of non-myelinated axons of >1 µm diameter at both time points (*, P < 0.05, Student's t test). Number of animals = 4. Number of quantified fibers per animal at 12 d = 50. Number of quantified fibers per animal at 1 mo = 150. Quantification of g-ratios 12 and 28 d after crush shows a significant increase in the g-ratio of the cRalA/B$^{-/-}$ mice compared to control littermates at both time points (*, P < 0.05, Student's t test, n = 4). Remak bundles, percentage of Remak bundles with polyaxonal pockets and number of C-fibers per Remak bundle, 12 d and 1 mo after injury, did not show any difference.

### RalGTPases control axial SC prolongation length and axial and radial lamellipodia formation in SCs

During development, SCs develop a cytoplasmic protrusion radially in relation to the main SC axis, similar to a giant lamellipodia in form and function (Nodari et al., 2007; Parkinson et al., 2008; Montani et al., 2014). This process is thought to be necessary for spiral membrane wrapping and myelin formation (Bunge et al., 1989; Feltri et al., 2002). In the first stages of nerve repair, myelinating and Remak SCs de-differentiate to repair SCs, adopting an elongated bipolar morphology. After myelin clearance, de-differentiated SCs align in columns inside the basal lamina to guide regenerated axons to reinnervate distal organs (Arthur-Farraj et al., 2012). To analyze the capacity of SCs lacking RalGTPases to extend and maintain axial and radial prolongations (major cytoplasmic process extensions of the cell) and lamellipodia (thin cytoplasmic sheets extended in the SCs; scheme in Fig. 6 A), we performed SC cultures from mice lacking both GTPases (Fig. 6 B). We then quantified the axial (at the two

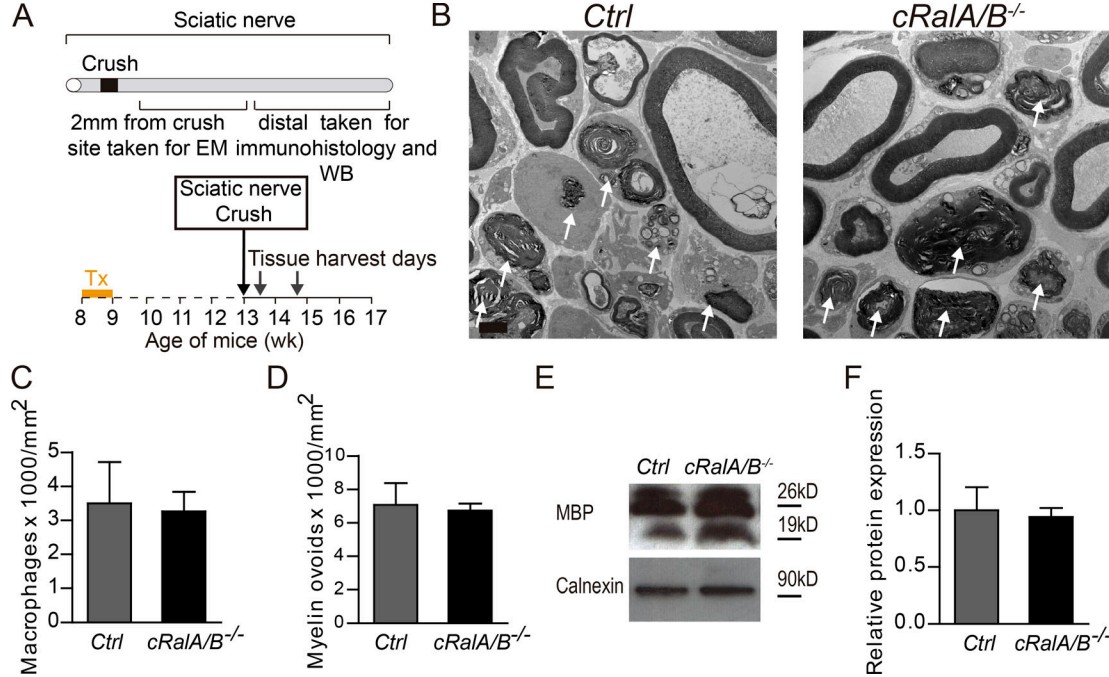

Figure 4. **RalGTPases are dispensable for macrophage recruitment and myelin clearance following peripheral nerve injury. (A)** Schematic representation of the nerve crush experiment at day 4. **(B)** Electron micrographs of transverse sections of sciatic nerve 4 d following crush. Arrows identify macrophages containing myelin ovoids. **(C and D)** Quantification of macrophage numbers (C) and myelin ovoid numbers (D). **(E)** Representative Western blot pictures for MBP expression in distal sciatic nerve homogenates 4 d after injury. **(F)** Quantification of relative expression of MBP showing no significant differences between mice. Data are presented as mean ± SEM. Scale bar, 2 µm.

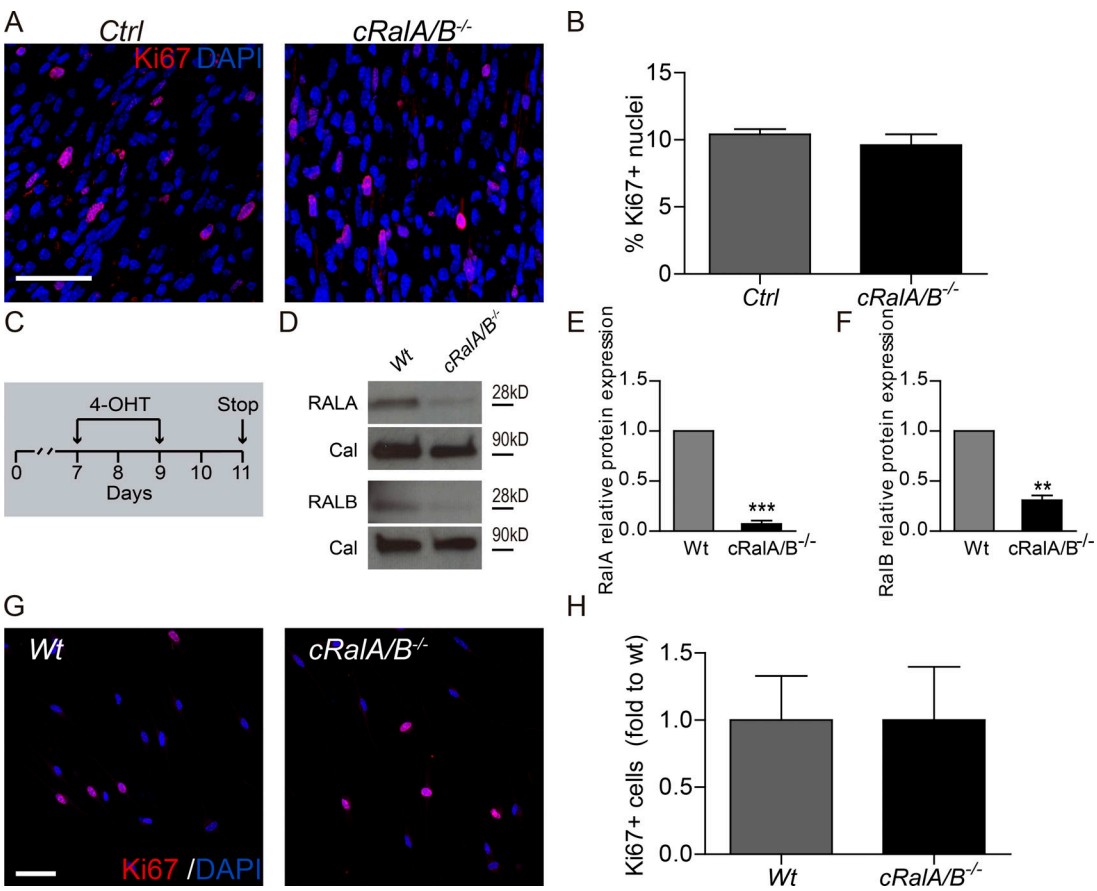

Figure 5. **RalGTPases are dispensable for SC proliferation after peripheral nerve injury and in vitro. (A)** Photomicrographs of longitudinal sciatic nerve sections of ctrl and *cRalA/B⁻/⁻* 12 d after crush. **(B)** Percentage of nuclei positive for Ki67 of sciatic nerves 12 d after crush. **(C)** Scheme of the 4-OHT administration on SCs for three consecutive days. SCs were left for 2 d with expansion medium. **(D)** Western blot to analyze the protein expression of RalGTPases on SC cultures quantified in E and F. **(G)** Representative images of SC cultures from WT and *cRalA/B⁻/⁻* mice. **(H)** Percentage of SCs positive for Ki67 in primary culture experiments. Proliferating nuclei are labeled with Ki67 (red) and nuclei are labeled with DAPI (blue). **, P < 0.01, ***, P < 0.001, Student's *t* test, *n* = 3 or 4. Data are presented as mean ± SEM. Scale bars, 50 µm.

extremities of the main cell axis) and radial (perpendicular to the main cell axis) SC prolongation length and lamellipodia numbers (Fig. 6 A).

We purified SCs from mouse sciatic nerve and then treated them with 4-OHT for 3 d. 1 d later they were cultured in new plates and synchronized with serum deprivation for 1 d in defined medium. After 4 h with expansion medium, we fixed the cells and analyzed the capability of mutant SCs to develop SC processes de novo (Fig. 6, B–E). We analyzed the effect of RalA and RalB ablation individually, and there was no significant difference in SC process formation using SC cultures derived from *cRalA* and *RalB⁻/⁻* mice compared with WT SCs (data not shown). When comparing SCs lacking both Ral GTPases (*cRalA/B⁻/⁻*) versus *RalB⁻/⁻*, we observed a significant decrease in both axial (Fig. 6 D) and radial (Fig. 6 E) lamellipodia formation in *cRalA/B⁻/⁻* mice compared with *RalB⁻/⁻*. Finally, we analyzed the length of both radial and axial prolongation (Fig. 6, F–H). Notably, although the length of radial SC prolongation was similar between groups, the length of axial prolongation was reduced in *cRalA/B⁻/⁻* SCs compared with *RalB⁻/⁻*. These results demonstrate the role of RalGTPases in (1) axial and radial lamellipodia formation and (2) axial prolongation length in SCs. This would

provide a cellular mechanism for the axon hypomyelination observed in vivo (Fig. 3).

**RalGTPases enhance axial and radial SC processes through different effectors**

RalGTPases can activate several downstream pathways through their downstream effectors (Bhattacharya et al., 2004; Lalli and Hall, 2005; Neel et al., 2011; Teodoro et al., 2013). To determine whether just one or more effectors are responsible for the alteration of SC processes observed in Fig. 6, we used viral vectors containing constitutive active Ral mutants that have lost the ability to bind to specific effector proteins and analyzed the capability to develop SC processes. WT rat SC primary cultures were transduced with lentivirus containing GFP as a control or GFP and Ral effector domain mutants. Cells were fixed 2 d after transduction, and axial and radial prolongations and lamellipodia number and axial and radial prolongation length were analyzed. To confirm that both viral vectors had the same efficiency of transduction, we took advantage of the fact that both CARalA and CARalB have an integrated Myc epitope, the levels of which were comparable in both conditions (Fig. S3 A). Then, we studied the effect of activated forms of both RalGTPases on SC

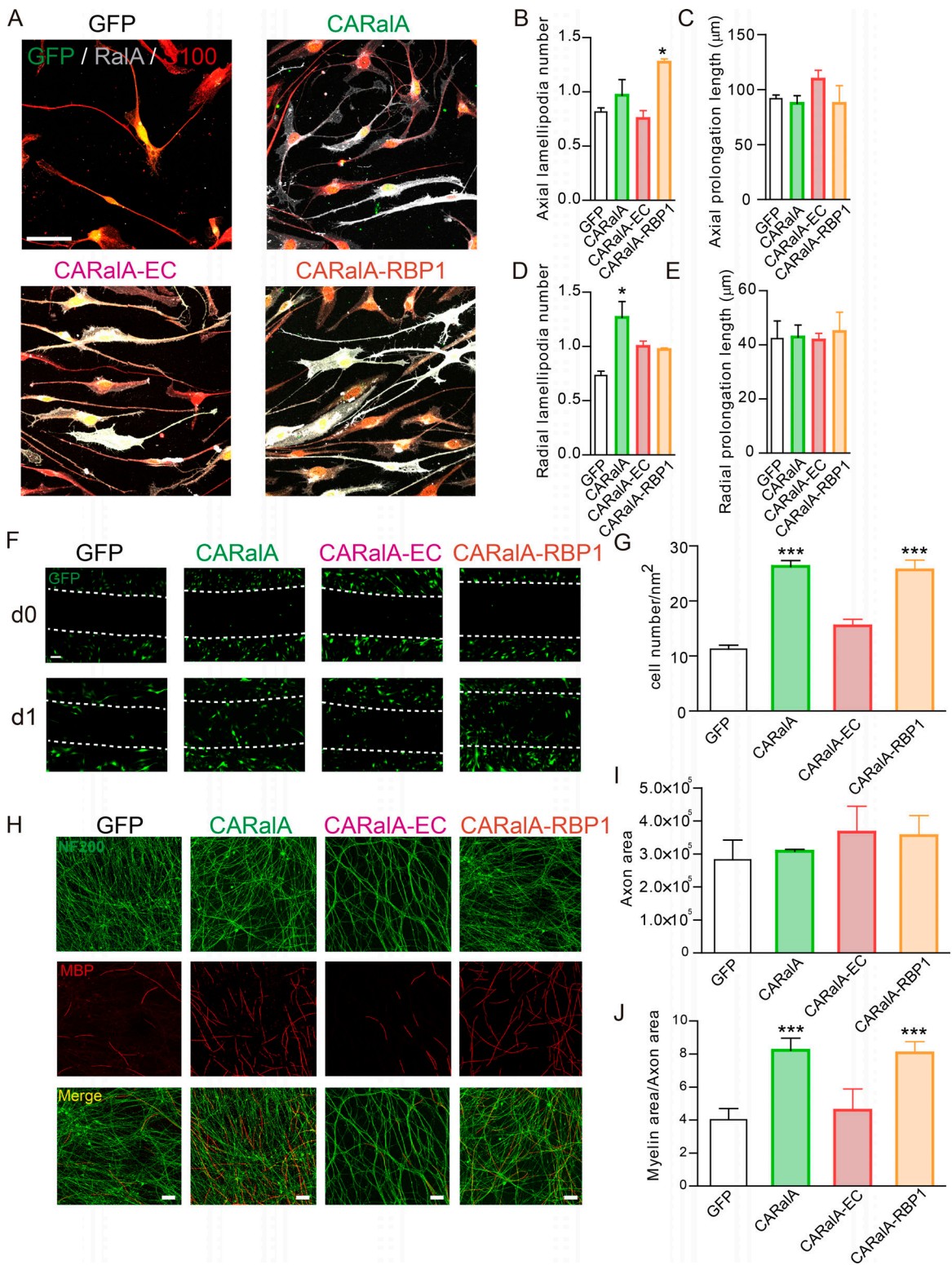

Figure 6. **Loss of RalGTPases results in disturbed SC prolongation and lamellipodia formation. (A)** Schematic representation of a SC showing axial (AP) and radial (RP) prolongations and axial (AL) and radial (RL) lamellipodia. **(B)** Scheme of the 4-OHT administration on SCs. **(C)** Images of SC cultures from *RalB⁻/⁻* (control) and *cRalA/B⁻/⁻* mice to study the formation of axial (white asterisks) and radial (yellow asterisks) SC lamellipodia and axial and radial SC prolongations in vitro. Axial (D) and radial (E) lamellipodia number per cell. **(F)** Representative images of three isolated SCs from *RalB⁻/⁻* and *cRalA/B⁻/⁻* mice to study the prolongation length. **(G and H)** Axial (G) and radial (H) prolongation average length. SC processes are stained with Talin (green) and nuclei with DAPI. Data are presented as mean ± SEM (*, P < 0.05, **, P < 0.01, Student's *t* test, *n* = 3 for each genotype). Scale bars, 50 µm.

Figure 7. **RalGTPases enhance SC radial processes number, SC migration, and axon myelination in an exocyst-dependent manner. (A)** Images of WT rat SC cultures transduced with lentivirus containing GFP as a control; CARalA, a constitutive active form of RalA; CARalA-EC, an active constitutive form of RalA unable to bind to exocyst; and CARalA-RBP1, a constitutive active form of RalA unable to bind to RalBP1. **(B–E)** Average of the number of axial (B) and radial (D) lamellipodia per cell; axial (C) and radial (E) prolongation length quantification. GFP is labeled in green, the SC marker S100 in red, and RalA and nuclei (with DAPI) in blue. SC processes number and length are quantified in S100 channel for all S100-positive cells. **(F and G)** The wound healing assay revealed differences in migration rate of SCs transduced with lentivirus containing the activated forms of RalGTPases. **(F)** Images of WT rat SC cultures for the wound healing assay. **(G)** Quantification of the average number of cells invading the wound area. Transduced cells are GFP⁺ (green). **(H)** Images of human iPSC-derived sensory neurons co-cultured with SCs transduced with lentivirus containing activated forms of RalA. **(I)** Viral transduction had no effect on axonal outgrowth.

**(J)** CARalA- and CARalA-RBP1–transduced SCs significantly promote myelination of human iPSC-derived sensory neurons. In contrast, CARalA-EC–transduced SCs had no effect on the myelination compared with GFP-transduced SCs. $n = 3$ for each virus condition and experiment. Data are presented as mean ± SEM (*, $P < 0.05$, ***, $P < 0.001$, one-way ANOVA, post hoc Tukey's test used to compare with GFP-transduced rat SCs). Scale bars, 50 µm.

sensory neurons are a good model to study the axon–SC relationship in order to initiate alignment, basal lamina formation, and myelination (Clark et al., 2017). To investigate the role of RalGTPases in myelination in this co-culture system, rat-derived SCs were transduced with activated forms of RalGTPases before the co-culture was performed. Viral SC transduction was well tolerated by iPSC-derived sensory neurons with no visible signs of cellular stress across different conditions (Fig. 7 H). Co-cultures performed with SCs transduced with CARalA had a significant increase in myelination compared with GFP-transduced SCs (Fig. 7, H and J). CARalB, however, did not enhance the myelination of iPSC-derived sensory neurons (Fig. S5, B and D). Interestingly, in co-cultures where SCs were transduced with CARalA-EC, the degree of myelination was not enhanced and was similar to that observed in control GFP virus–transduced SCs (Fig. 7, H and J). However, CARalA-RBP1–transduced SCs were able to increase myelination versus control similarly to CARalA transduced SCs (Fig. 7, H and J). We did not observe any significant difference in axon area with any of the activated forms of RalGTPases (Fig. 7, H and I). In summary, these results suggest that activation of exocysts by RalA is important for SC migration and myelination, and gain of function of RalB does not enhance these SC functions.

### RaGTPase is required for radial lamellipodia formation in an exocyst-dependent manner

To demonstrate that SC process formation and extension are dependent of RalGTPases, we transduced SCs derived from *cRalA/B*$^{-/-}$ mice with lentivirus containing activated forms of RalGTPases (Fig. 8). As expected, SCs derived from *cRalA/B*$^{-/-}$ mice transduced with CARalA or CARalB mutants recovered SC process formation and extension observed in control SCs transduced with Lv-GFP (Fig. 8). CARalA-EC did not, however, rescue the radial lamellipodia number per cell to the levels observed in control SCs (Fig. 8 D) but rescued axial lamellipodia (Fig. 8 B) and prolongation lengths (Fig. 8, C and E). These results demonstrate that SC processes are regulated by RalGTPases and that exocyst activation mediated by RalGTPases is important for radial lamellipodia formation in SCs.

## Discussion

In this study, we have shown that RalGTPase signaling within SCs contributes to optimal nerve repair. RalGTPases were required for the formation and extension of both axial and radial processes of SCs. Enhanced RalGTPase signaling could enhance both SC migration and myelination. These effects were dependent on interaction with the exocyst complex.

### Ral GTPase signaling contributes to effective nerve repair

Our data in transgenic animals lacking both GTPases demonstrated that RalGTPases were dispensable for the maintenance of

axon integrity and the myelin sheath in the naive state in adulthood. However, the absence of RalGTPase signaling in SCs seriously impaired the regenerative process following nerve injury: we found that RalGTPases played an important role in axon remyelination, the reinnervation of distal organs, and restoration of motor function after injury. There was significant redundancy in that ablation of RalA or RalB individually had no impact on nerve repair. Such redundancy has previously been observed in studies of the role of RalA and B in development and tumorigenesis (Peschard et al., 2012). Interestingly, when we examined the expression of RalGTPases in the distal nerve stump of WT mice following injury, we observed a complex pattern with decreased expression of RalA and increased expression of RalB. When assessing the activated forms of RalA and B, however, there was a trend for enhanced activation of RalA and significantly reduced activation of RalB following nerve injury. It has been previously noted that changes in the expression are not necessarily correlated with activation status of RalGTPases (Peschard et al., 2012). Despite the differential expression between RalA and B, there was clear redundancy at a functional level when individually ablated.

The regeneration process following nerve injury includes dedifferentiation of SCs to a regenerative state followed by redifferentiation, association with regrowing axons, and formation of a myelin sheath (Chen et al., 2007; Arthur-Farraj et al., 2012). There were morphological differences in the outcome of nerve repair following sciatic nerve crush in *cRalA/B*$^{-/-}$ mice versus ctrl, and in particular RalGTPases were required for axon remyelination. In *cRalA/B*$^{-/-}$ mice, we observed that (1) the density of axons of >1 µm that were unmyelinated was increased and (2) those axons that were myelinated had a thinner myelin compared with ctrl mice. These features were observed at 12 d and 1 mo after injury. In addition to this deficit in radial myelination, we found that in the absence of RalGTPases signaling internodal length was shorter. In contrast, we did not observe any changes in nonmyelinating SCs and their associated C-fibers: the number of C-fibers per Remak bundle and the number of Remak bundles in the injured nerve did not change after RalGTPases ablation.

We also observed a decrease in reinnervation of the neuromuscular junction of the gastrocnemius muscle 1 mo after injury in *cRalA/B*$^{-/-}$. In these animals, RalB is absent in regenerating axons as well as SCs. However, the fact that we did not observe any difference in neuromuscular junction reinnervation after injury in *RalB*$^{-/-}$ mice suggests that the likely explanation for this phenomenon is impaired RalGTPase signaling in SCs. One of the steps of SC repair response to injury is the formation of regeneration tracks (Chen et al., 2007). After injury, SCs change to a more bipolar morphology, they elongate and align to form a SC column named a band of Bungner, and SCs also form "bridges" at points where there is a loss of nerve integrity (particularly relevant to axotomy; Parrinello et al., 2010). These

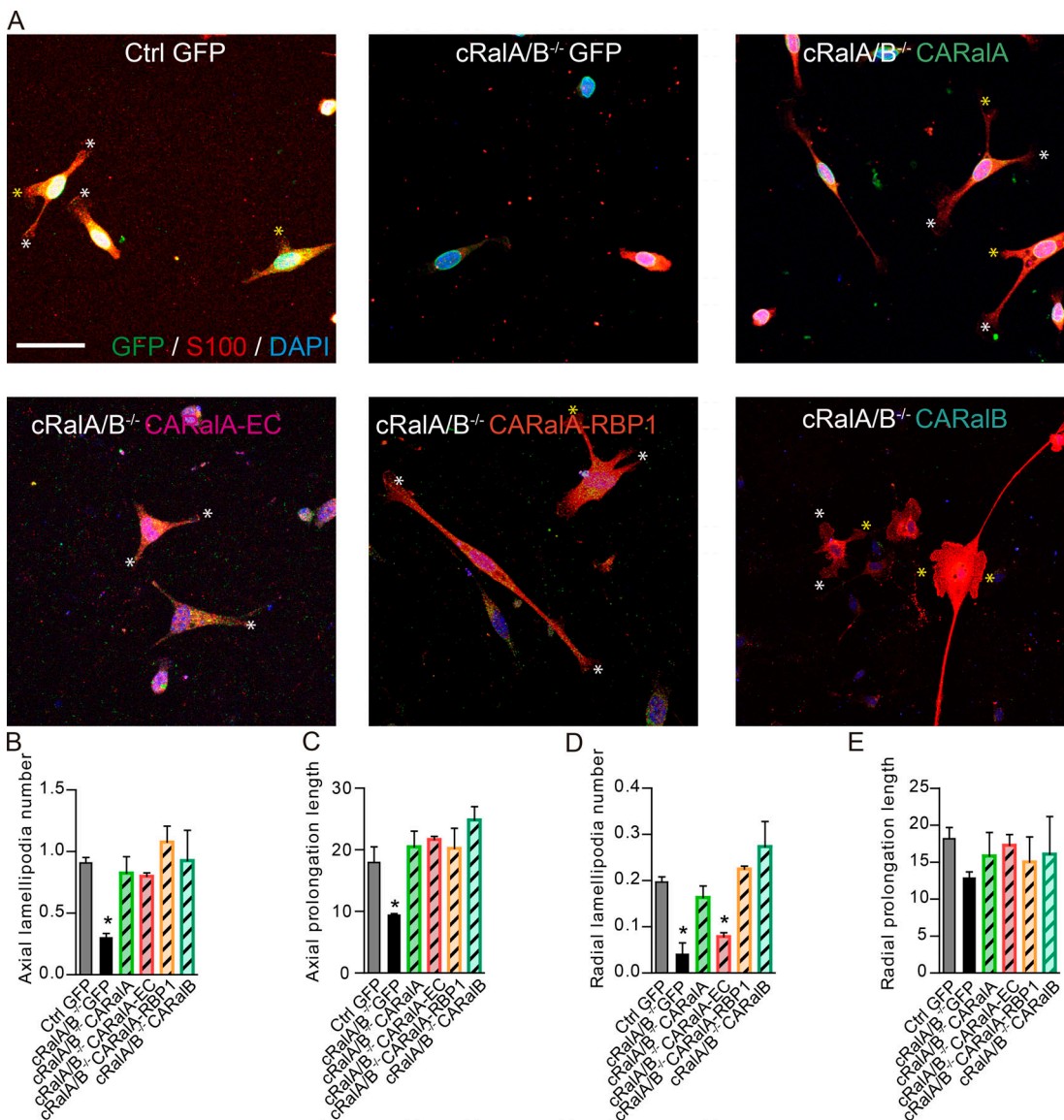

Figure 8. **Activated RalGTPases rescue SC process formation on *cRalA/B*⁻/⁻-derived SCs. (A)** Images of SC cultures from WT rat transduced with GFP as a control (Ctrl GFP) and from *cRalA/B*⁻/⁻ mice transduced with lentivirus containing GFP as a control; CARalA, a constitutive active form of RalA; CARalA-EC, a constitutive active form of RalA unable to bind to exocyst, and CARalA-RBP1, a constitutive active form of RalA unable to bind to RalBP1. **(B–E)** Average of the number of axial (white asterisks; B) and radial (yellow asterisks; D) lamellipodia per cell; and axial (C) and radial (E) prolongation length quantification. SC processes number and length are quantified in S100 channel for all S100-positive cells. GFP, green; S100, red; DAPI, blue. n = 3 for each virus condition and experiment. Data are presented as mean ± SEM (*, P < 0.05, one-way ANOVA, post hoc Tukey's test used to compare with GFP-transduced control SCs). Scale bar, 50 µm.

columns form tracks along the distal regenerating nerve and provide guidance to axons to achieve their distal organs. We have observed a trend in the reduction of SC axial prolongations and a decrease in their elongation in vitro in SCs from *cRalA/B*⁻/⁻ (discussed below), which could impede these close SC–axon interactions. Terminal SCs within the muscle have also been shown to be critical in synaptic reinnervation by motor axons (Balice-Gordon, 1996; Kang et al., 2014). As well as physical interactions, it is also possible that RalGTPase-deficient SCs produce fewer growth factors for axons (Jessen and Mirsky, 2016).

The functional recovery from nerve injury was impaired in *cRalA/B*⁻/⁻ mice. The SFI is a measure of motor functional

recovery: we noted significant delay in the recovery of SFI following nerve crush in the absence of both RalGTPases. Interestingly, the recovery was delayed rather than prevented, and by 28 d after injury, there was no longer a significant difference. Although remyelination and target reinnervation had not completely normalized at this stage, they were clearly sufficient for reasonable performance in this assay. We also tested sensory function by using the pin prick test to determine cutaneous reinnervation of the hindpaw. This was not significantly different in *cRalA/B*⁻/⁻ mice, consistent with the fact that we did not observe any morphological alterations in Remak bundle structure in these mice.

We examined the proliferation of SCs as well as myelin clearance in case these could provide insight into defective nerve repair in *cRalA/B^{-/-}* mice. Nerve injury and repair is associated with two phases of SC proliferation: the first is due to loss of axonal contact and peaks around 3 d after injury (Carroll et al., 1997; Bradley et al., 1998; Stoll and Müller, 1999; Griffin and Thompson, 2008); the second occurs as axons enter the denervated nerve stump (Pellegrino and Spencer, 1985). RalGTPases promote cell proliferation in some models of cancer (Gentry et al., 2014) and, of particular relevance, in some lines of malignant peripheral nerve sheath tumors (Bodempudi et al., 2009). Ral GTPases have also been shown to interact with the small GTPase cdc42, which activates SC proliferation during development (Benninger et al., 2007). Active (GTP-ligated) Ral associates with RalBP1. RalBP1 is a GTPase-activating protein for Cdc42, which it deactivates (Matsubara et al., 1997). We therefore investigated the role of RalGTPases in the proliferation of SCs following nerve injury. Cellular proliferation within the distal nerve stump at 12 d after injury was not altered in the absence of both RalGTPases. Furthermore, the proliferation of SCs in vitro did not change in the absence of RalGTPase.

A further early event in peripheral nerve repair is the clearance of myelin debris (Chen et al., 2007). Both phagocytosis of myelin by macrophages as well as phagocytosis and autophagy of myelin by SCs contribute to myelin clearance (Gomez-Sanchez et al., 2015; Velasquez et al., 2018). We did not find any change in myelin clearance during the regeneration process in vivo in *cRalA/B^{-/-}* mice.

### RalGTPases are required for the SC processes formation and extension, SC migration, and axon myelination

The dynamic interactions between SCs and axons in the distal nerve stump are critical for successful nerve repair (Arthur-Farraj et al., 2012). This requires significant morphological changes in SCs; for instance, repair SCs are reportedly two- to threefold longer than quiescent myelin and Remak SCs (Gomez-Sanchez et al., 2017). Other small GTPases, particularly of the Rho family, such as Rac1 and cdc42, have been shown to be critical for SC process formation and myelination during development and after repair (Nodari et al., 2007; Stendel et al., 2007; Park and Feltri, 2011; Guo et al., 2012, 2013). We therefore investigated the role of RalGTPases on SC process formation. Our in vitro assays demonstrated that the ablation of RalA and RalB GTPase in SCs reduced their ability to form normal axial and radial lamellae. As well as studying loss of function, we interrogated the effect of enhancing the activity of individual RalGTPase on the SC processes formation, migration, and myelination. The active forms of RalA and B enhanced radial lamellipodia formation. The cellular phenotype of SCs transduced with active forms of RalGTPases was not, however, the exact opposite of RalGTPases—null cells in that axial lamellipodia number and axial prolongation length were not enhanced. This difference may reflect the "set point" of GTPase signaling within the cell, and the stimulus-response function may differ between axial and radial processes (Pankov et al., 2005; Nodari et al., 2007); there may also be differences in the role of RalGTPases for maintenance of SC processes versus increasing the length or

enhancing formation of new processes. There were some differences in comparing the effects of active RalA versus RalB: both GTPases increased radial lamellae; however, only RalA increased radial prolongation numbers. The directed migration and elongation of cells depend on their ability to produce lamellipodia (Pankov et al., 2005). Furthermore, radial lamellipodia formation by SCs is required for initial axon ensheathment and then wrapping during the process of myelination (Bunge et al., 1989; Nodari et al., 2007; Montani et al., 2014). Consistent with the effect of RalA on SC process formation, we observed an increase in SC migration by the activated form of RalA. We also found that the activated form of RalA enhanced myelination in an in vitro co-culture assay. In contrast to RalA, the active form of RalB did not enhance SC migration or myelination in vitro, consistent with the fact that RalB was less effective at promoting process formation.

An important effector of RalGTPases is interaction with and activation of the exocyst complex. We therefore studied a RalGTPase mutant that is unable to attach and activate exocysts. This failed to increase (1) migration rate, (2) axon myelination, and (3) rescue of the SC morphology observed in SCs derived from *cRalA/B^{-/-}* mice, suggesting a specific role for the exocyst complex in mediating the effects of Ral on SC morphology and function. This Ral-exocyst signaling may regulate process formation and ultimately migration and myelination via a number of mechanisms, including (1) the targeted delivery proteins such as P0, myelin-associated glycoprotein, or laminin by exocytosis (Chen et al., 2012); (2) addition of membrane to plasma membrane (Trapp et al., 1995); or (3) the polarization of SCs, which is known to be a critical for myelination (Lewallen et al., 2011; Shen et al., 2014). RalGTPases also interact with RalBP1; however, our in vitro experiments using forms of RalGTPases, which are unable to activate RalBP1, demonstrated that this does not appear to be an effector by which RalGTPase enhances process outgrowth, SC migration, and myelination.

In conclusion, we have identified that RalGTPase activity in SCs contributes to effective nerve repair and specifically for the axial and radial elaboration of myelin by SCs as they interact with regenerating axons. We found that RalGTPases are required for the formation of SC processes through their interaction with the exocyst complex. This provides a further example of the critical role of SCs in coordinating effective nerve repair, and such signaling could potentially be targeted therapeutically.

## Materials and methods

### Transgenic mice

All procedures were performed in accordance with UK Home Office regulations (Mice Scientific Procedures Act, 1986). *RalA^{fl/fl}* and *RalB^{-/-}* mice were provided by C. Marshall (Institute of Cancer Research, London, UK) and the generation and genotyping of these mice has previously been described (Peschard et al., 2012). *cRalA* and *cRalA/B^{-/-}* were bred by crossing *PLPCreER^{T2}* (Doerflinger et al., 2003) and JAX-005975 (http://jaxmice.jax.org/strain/005975.html) with *RalA^{fl/fl}* and *RalA^{fl/fl}; RalB^{-/-}*, respectively. Conditional ablation of RalA in SCs (here called *cRalA*) and conditional ablation of RalA in SCs of RalB KO

mice (here called *cRalA/B*$^{-/-}$) were generated by administering tamoxifen (0.25 mg/g body weight in corn oil; T5648; Sigma-Aldrich) by oral gavage for five consecutive days to 8-wk-old *PLPCreER*$^{T2}$:*RalA*$^{fl/fl}$ (*cRalA*) mice or *PLPCreER*$^{T2}$:*RalA*$^{fl/fl}$:*RalB*$^{-/-}$ (*cRalA/B*$^{-/-}$) mice, respectively. Tamoxifen was administered 4 wk before surgery. For both experiments, one type of control mice was used for comparison: tamoxifen control mice (ctrl), which are tamoxifen-treated Cre-negative littermates (i.e., *Ral*$^{fl/fl}$ or *RalA*$^{fl/fl}$:*RalB*$^{-/-}$, respectively). To detect the PLPCreER$^{T2}$ construct, PCR of genomic DNA using the following primers was performed: 5′-AGGTGGACCTGATCATGGAG-3′ and 5′-ATACCGGAGATCATGCAAGC-3′ (performed as follows: heating at 94°C for 3 min, 40 cycles of 94°C for 30 s, 52°C for 30 s, and 72°C for 30 s, followed by a final extension at 72°C for 8 min). In *PLPCreER*$^{T2}$ mice, a tamoxifen-inducible form of Cre recombinase is expressed in SCs by the PLP promoter (Doerflinger et al., 2003). For experiments comparing WT and *RalB*$^{-/-}$ mice, heterozygous mice for RalB mutant allele (*RalB*$^{+/-}$) were bred together, and WT and *RalB*$^{-/-}$ littermates were used for experimental procedures. Wherever possible, an equal number of mice of each gender in each experimental group were included. This study conforms to the Animal Research: Reporting of In Vivo Experiments guidelines. Each scientific question in which statistical analysis was performed was addressed with a sample size chosen based on a power calculation using historical data relating to SFI responses (α error of 0.05 and a power of 80%). This calculation was suggesting a minimal cohort size of 3–6 for anatomical and 8–12 for behavioral studies. The sample size for each experiment is indicated in the corresponding figure legend. Samples were allocated into experimental groups based on their genotype. The experimenter was blinded to the group allocation during the experiment and/or when assessing the outcome.

### Surgery
Surgery was performed in 13-wk-old mice (in those mice that had been treated with tamoxifen, this was 4 wk after treatment). Under sterile conditions and inhalational anesthesia, the left sciatic nerve was exposed and crushed twice, in two different directions for 30 s each time, with fine forceps. In all cases, the lesion site was kept constant, and the wound was closed with 6–0 sutures and disinfected. Post-surgery analgesia drugs were given to the mice, and autotomy and weight loss were monitored daily for the first week after surgery and three times per week later on.

### Lentivirus production
All Ral mutant constructs were a gift from G. Lalli, Kings College London, London, UK (Lalli and Hall, 2005). The subcloning of Ral constructs to viral vectors, viral packaging, CsCl purification, and viral genome titration were performed by Creative-Biogene. All lentivirus was made with a ZsGreen1 protein linked to the specific RalGTPase construct RalA72L (CARalA), CARalA-EC, CARalA-RBP1, and RalB23V (CARalB) with an IRES sequence as a linker. Primary rat and mice SCs were transduced 1 wk after the culture was established onto coverslips. For all cell transductions, the stock virus was diluted in DMEM to obtain the correct multiplicity of infection of 5–10 based on the number of cells per coverslip.

### Mouse and rat SC purification
For mice SC purification, sciatic nerves from pups on postanal day 8–10 were dissected and placed in HBSS on ice. The epineurium and superfluous connective tissue were removed using forceps. Desheathed nerves were then cut into 5-mm segments and were enzymatically digested with a collagenase (3 mg/ml; Sigma-Aldrich) and dispase II protease (2.5 mg/ml; Sigma-Aldrich) incubation for 1 h at 37°C. Nerves were gently triturated using a P1000 and then a P200 pipette tip and plated onto poly-D-lysine (Sigma-Aldrich)/laminin (R&D Systems)–coated coverslips in SC expansion medium (DMEM without L-glutamine and with D-valine to avoid fibroblast contamination; custom-made by Cell Culture Technologies), 10% horse serum (Thermo Fisher Scientific), 200 ng/ml NRG1-β1 EGF domain (R&D Systems), 10 ng/ml NGF (recombinant-murine; Peprotech), 4 μg/ml forskolin (Sigma-Aldrich), and GlutaMAX (Invitrogen) for 1 wk until the myelin and other cell debris were completely removed, changing medium three times that week. Mouse SC cultures were then treated with 2 μM 4-OHT for 1 d and the day after with 1 μM for two consecutive days. SCs were either left for 2 d with expansion medium or replated and left in serum-free medium for 1 d followed by 4 h of expansion medium. In experiments that needed the lentivirus transduction in mice SCs, this was performed on the second day of 4-OHT treatment (Fig. 8 A). For serum-free conditions, SCs were cultured in defined media (DMEM/F12 media supplemented with 10 mg/liter forskolin, 10 μM putrescine, 20 nM progesterone, 30 nM sodium selenite, and GlutaMAX).

Rat SCs were purified as described by Clark et al. (2017). SCs were harvested from the sciatic nerve and brachial plexus of postnatal day 3 or 4 rats. Nerves were enzymatically digested with a collagenase (3 mg/ml; Sigma-Aldrich) and dispase II protease (2.5 mg/ml; Sigma-Aldrich) incubation for 1 h at 37°C. Nerves were gently triturated using a glass-blown pipette and plated onto poly-D-lysine/laminin-coated plastic in DMEM/F12 (Thermo Fisher Scientific) and 10% FBS (Thermo Fisher Scientific). From the day after, SCs were kept in expansion medium (DMEM/F12, 10% FBS, 10 mM NRG1-β1 EGF domain [R&D Systems], 10 ng/ml NGF [recombinant-murine; Peprotech], and 4 μM forskolin [Sigma-Aldrich]). Cells were serially treated with 5–10 μM araC to eliminate fibroblasts. After approximately four passages, each time doubling the growing area, cells were frozen in a Mr. Frosty freezing container (Thermo Fisher Scientific) and stored in liquid nitrogen. The cells were transduced with lentivirus at multiplicity of infection 5–10 and fixed 3–4 d after transduction.

### SC migration
Rat SCs were transduced with lentivirus at multiplicity of infection 10 and plated in 4-well plates at a concentration of $5 \times 10^4$ cell/well, in triplicate for each condition. After reaching the confluence, a gentle scratch in the well was done with a P200 tip without touching the poly-D-lysine/laminin coating, and marks were done on the plate in order to easily identify the area.

## Establishing the myelinating co-cultures

iPSC generation, maintenance, and differentiation to neuronal culture was performed as described by Clark et al. (2017). To establish the co-cultures, the medium on the neuronal cultures was changed from N2 to SC basal medium (DMEM/F12 [Thermo Fisher Scientific], 5 mg/ml insulin [Sigma-Aldrich], 100 mg/ml transferrin [Millipore], 25 ng/ml NGF [recombinant-human; Peprotech], 25 ng/ml selenium [Sigma-Aldrich], 25 ng/ml thyroxine [Sigma-Aldrich], 30 ng/ml progesterone [Sigma-Aldrich], 25 ng/ml triiodothyronine[Sigma-Aldrich], and 8 mg/ml putrescine [Sigma-Aldrich]). Transduced SCs (25,000) in a 25-µl droplet were slowly pipetted onto a single coverslips containing the iPSC-derived sensory neurons, with care taken not to touch the coverslips with the pipette tip. The SC droplet was evenly distributed over the coverslip. Cultures were carefully put back into the incubator to allow SC adherence. The culture medium was changed 2 d later. At this stage, abundant SCs could be observed adhered to the coverslip. An additional SC basal medium change was performed 4 d after adding the SCs. Myelination was induced 1 wk after SC addition by exposing the cells to myelination medium (N2 medium, 1:300 phenol-free Matrigel [Scientific Laboratory Supplies], 5% charcoal-stripped FBS [Thermo Fisher Scientific], 25 ng/ml NGF [recombinant-human; Peprotech], and 50 mg/ml ascorbic acid [Sigma-Aldrich]). Medium changes were performed twice weekly from then on. The myelin quantification by immunocytochemistry was performed 4 wk after transduced SC addition as reported by Clark et al. (2017).

## Western blotting

Tissues and primary SC cultures were homogenized in NP-40 lysis buffer (20 mM Tris, pH 8, 137 mM sodium chloride, 10% glycerol, 1% NP-40, 2 mM EDTA, 20 µM leupeptin, 1 mM PMSF, 1 mM sodium orthovanadate [all from Sigma-Aldrich], 5 mM sodium fluoride [BDH], and protease inhibitor cocktail [Roche]) with a Dounce (Sigma-Aldrich) and a syringe, respectively. The lysates were kept rotating at 4°C for 90 min, then spun at 13,000 rpm at 4°C for 15 min, and the protein concentration of supernatant was determined using a BCA Protein Assay kit (23227; Thermo Fisher Scientific). 10–30 µg of protein was mixed with SDS gel sample buffer, electrophoresed on 8–14% Mini-PROTEAN TGX Precast Protein Gels (Bio-Rad), and transferred to Immobilon-P Membrane, polyvinylidene fluoride, 0.45-µm membranes (Millipore). Blocking was done in 3% BSA (Europa Bioproducts) and PBS. Primary antibodies used were αms RalA (BD Transduction Laboratories) at a dilution of 1 to 1,000, αrt RalB (R&D Systems) at a dilution of 1 to 500, αms Myc (Cell Signaling) at a dilution of 1 to 500, αrb calnexin (Enzo Life Science) at a dilution of 1 to 1,000, and αms GAPDH (ab8245; Abcam) at a dilution of 1 to 1,000. Secondary antibodies used were anti-rabbit IgG HRP linked (NA9340V; GE Healthcare) and anti-mouse IgG HRP linked (NA931V; GE Healthcare) at 1 to 20,000, and anti-rat (DAKO) at 1 to 5,000. The ECL prime Western blotting detection system (GE Healthcare) was used to visualize the immunoreactive band on chemiluminescence film (GE Healthcare). For quantification and analysis, the intensity of specific bands was quantified using ImageJ. Each band detected by RalA and RalB was normalized against loading control calnexin or GAPDH correspondingly for analysis.

## Immunoprecipitation and RalGTPase activity

RalGTPase activity in vivo was assessed using the Ral activation assay (New East Bioscience) following product guidance with some modifications. Homogenates of a pool of two sciatic nerves from two different animals, to generate each data point, with immunoprecipitation Lysis Buffer (50 mM Tris-HCl, 125 mM NaCl, 1% NP-40, 0.1% SDS, and anti-proteases), were incubated for 1 h at RT with magnetic beads (Bio-Rad) previously incubated with 1 µg of anti-active Ral antibody (New East Bioscience) for 10 min. To elute the antigen from the bead-antibody complex, the beads were incubated 10 min with 40 µl of 1× Laemmli buffer at 70°C. Immunoprecipitation products were then loaded into 12% Mini-PROTEAN TGX Precast Protein Gels (Bio-Rad) and transferred to an Immobilon-P Membrane, polyvinylidene fluoride, 0.45-µm membranes (Millipore). Finally, membranes were incubated with antibodies against anti-RalA or -RalB, as described in the Western blotting section, to detect activated forms of RalA and RalB, respectively.

## Immunohistochemistry

Mice were deeply anaesthetized with pentobarbitone and transcardially perfused with either 5 ml saline or 5 ml saline followed by 25 ml paraformaldehyde (4% in 0.1 M phosphate buffer [PB]). Gastrocnemius muscle and sciatic nerves were post-fixed in paraformaldehyde (4% in 0.1 M PB) for 30 min (muscle) and 1 h (sciatic nerve) and then transferred to 30% sucrose for 3 d. Tissue was then mounted in optimal cutting temperature embedding compound on dry ice and stored at −80°C. Sciatic nerves for nerve teasing were post-fixed in paraformaldehyde (4% in 0.1 M PB) for 30 min and stored in PBS, 0.1% NaAz at 4°C.

For muscle staining, longitudinal sections of gastrocnemius were cut on a cryostat in 100 µm sections and transferred into a 24-well plate and stored in PBS containing 0.1% NaAz followed by incubation with antibodies against the presynaptic marker αms SV2 (1 in 100; DSHB) and axon marker αms 2H3 (1 in 50; DSHB) and tetramethyl-rhodamin–conjugated α-bungarotoxin (1 in 1,000; Invitrogen) to stain the post-synaptic acetylcholine receptors. Sections were washed three times with PBS, followed by incubation in blocking solution (PBS, 5% normal donkey serum, 1% BSA, 1% DMSO, 0.5% milk protein, 0.3% Triton X-100, and 0.1% NaAz) for 30 min at RT. Immunoreaction with primary antibodies 2H3, SV2, and MBP was performed by incubation overnight at RT in PBS and 0.2% Triton X-100. After washing in PBS, corresponding secondary antibodies and α-bungarotoxin were diluted in PBS and 0.2% Triton X-100 and incubated for 2.5 h at RT. Sections were washed, transferred to slides, and mounted with Vectashield mounting medium for fluorescence (VECTOR). The total number of neuromuscular junctions and the number that were innervated by axons were counted using ImageJ.

For Ki67 staining, longitudinal sections of sciatic nerve were cut on a freezing microtome in 14 µm sections. Sections were stained for the proliferation marker Ki67. Before Ki67 staining,

nerves were incubated with an antigen retrieval buffer (10 mM citrate buffer, pH 6.2) overnight at 60°C. After washing, the sections were incubated with primary antibody αrb Ki67 (Abcam) dilution 1 to 500 overnight at RT in PBS and 0.2% Triton X-100. Slides were washed three times with PBS for 5 min each, incubated in the corresponding secondary antibody containing DAPI (1 to 50; Invitrogen), and diluted in antibody buffer for 2.5 h at RT. Slides were washed three times with PBS for 5 min and mounted with Vectashield mounting medium for fluorescence (VECTOR) and examined with the confocal Zeiss LSM 700 laser scanning microscope.

For nerve teasing in order to measure internodal length, sciatic nerves were incubated with 0.5% osmium tetroxide (TAAB) for 2 h followed by three washes of 5 min each with 25% EtOH, 50% EtOH, and 75% EtOH. Nerves were stored in 50% glycerol and 50% of 70% EtOH and teased in glycerol. Sciatic nerves were teased under a microscope. The perineurium was removed from the fascicles using fine forceps and sharp needles. Nerve bundles were divided until single fibers could be teased out. Slides were covered with vectashield mounting medium for fluorescence (VECTOR) and analyzed at the confocal microscope.

For immunostaining of teased nerves, sciatic nerves were fixed in 4% PFA for 30 min. After perineurium removal, the sciatic nerve was teased as described above. Fibers were then incubated with primary antibodies in blocking buffer (1% BSA in PBS) overnight at 4°C. Incubation with secondary antibodies was done in blocking buffer for 1 h at RT. Primary antibodies used were monoclonal αms RalA (610222BD), 1 to 50, and αrb S100 (Z0311; DAKO), 1 to 300.

### Immunocytochemistry

Primary rat and mice SCs and human iPSC-derived sensory neurons–SCs co-culture were fixed in 4% PFA for 10 min at RT. Primary cells were then permeabilized with 0.2% Triton X-100 for 5 min at RT, whereas the co-cultures were permeabilized with cold MeOH for 20 min on ice. Cells were incubated with primary antibody in blocking buffer (1% BSA in PBS) overnight at 4°C, followed by incubation with secondary antibodies in blocking buffer for 2.5 h at RT. Primary antibodies used in immunocytochemistry were monoclonal αms RalA (610222; BD Biosciences), 1 to 50; αrb S100 (Z0311; DAKO), 1 to 300; αrb Ki67 (ab16667; Abcam), 1 to 500; αck GFP (ab13970; Abcam), 1 to 200; αgt Talin (sc-7534; Santa Cruz Biotechnology), 1 to 50; αms Myc (2276s; Cell Signaling Technology), 1 to 500; αms neurofilament heavy chain (ab7795; Abcam), 1 to 10,000; αrt MBP (ab7349; Abcam), 1 to 400; and αrb S100 (GA504; DAKO), 1 to 300.

### Image acquisition and analysis

Images were taken with the Axio Camera of the Zeiss Imager Z1 microscope and using Carl Zeiss ZEN SP1 black software.

Reinnervation of the gastrocnemius was quantified by taking ~10 pictures of each animal muscle at 20× magnification with a Z-stack through the entire thickness of the muscle section. Using ImageJ software, the ratio of the total number of innervated endplates and the total number of endplates was used for the percentage of reinnervated endplates. The ratio of reinnervated endplates that are myelinated and the total number of re-innervated endplates was used for the percentage of myelinated endplates.

In vivo cell proliferation was quantified using ImageJ in approximately five pictures at 20× magnification with a Z-stack of each nerve in which all Ki67 positive nuclei were counted and normalized by the total number of nuclei (750 per animal), indicating the percentage of proliferating cells.

In vitro SC proliferation was quantified in approximately five random tile scan pictures of each culture (300 cells per culture) using ImageJ.

For internodal distance, 70–100 lengths per animal were measured using the ImageJ ROI Manager software.

For the study of the SC processes extension and formation, primary rat and mice SC cultures were used. The number and length of all radial and axial prolongations (all major cytoplasmic process extensions of the cell) and the number of radial and axial lamellipodia (thin cytoplasmic sheets extended at the edge of an axial prolongation or in radial plane of the SCs) were analyzed. SC processes at the two extremities of the main cell axis were considered axial prolongation/lamellipodia, and the ones opposite to the main cell axis were considered radial prolongation/lamellipodia (for more clarification, follow the scheme in Fig. 5 A). For each independent mouse/rat-derived SC primary culture, three coverslips were plated and analyzed. 30–40 SCs were randomly chosen per coverslip to count prolongations and lamellipodia per cell and measure the length of the prolongations.

For the quantification of transduction efficiency, primary mouse and rat SCs were used. The percentage of S100/GFP-positive cells was calculated analyzing four images for each condition. Images were taken at 10× magnification for mouse SCs and 20× magnification for rat SCs. The intensity of GFP fluorescence in single cells was calculated analyzing 50 cells for each condition. Intensity was measured using the ImageJ software in pictures taken with exactly the same setting conditions.

For the quantification of the migration area, representative pictures were taken immediately after the scratch and 24 h later with the fluorescence microscope. Taking advantage of the ZsGreen1 protein linked to each RalGTPase construct, migrating SCs were quantified using the ImageJ software, and each count was normalized for the area of the scratch in square millimeters.

For the quantification of myelination in co-cultures, Zen Black Software (Zeiss) was used to create a digital diamond shape of 13 points (2-mm horizontal/vertical distance between each point) that was centered on each coverslip. A tile scan (each 1,792 × 1,792 µm in size, using a 20× objective) was taken at each point, resulting in 13 nonoverlapping images that equally sampled all quadrants of the coverslip (31.4% of the total area). Images were processed and analyzed using ImageJ software (version 1.48J). A condition-blinded investigator then created binary images for each channel (MBP and NF200 [eurofilament heavy chain]) by adjusting the threshold to acquire the maximum signal-to-noise ratio, and area measurements for each channel were recorded. The proportion of axonal area covered by myelin was then calculated as reported in Clark et al. (2017).

## Electron microscopy

Sciatic nerves were dissected, and a section 2–3 mm distal to the crush site on the injured side and an equivalent level on the uninjured side were processed for EM. Sciatic nerves were post-fixed with 4% PFA and 3% glutaraldehyde in 0.1 M PB and processed as previously described (Fricker et al., 2009). Nerves were osmicated with 1.5% osmium/0.2 M PB for 90 min, dehydrated, and embedded in epoxy resin (TAAB). Ultrathin sections (90 nm) were cut using a Diatome diamond knife on the Leica UC7 ultramicrotome and mounted onto 100 mesh Cu grids coated with formvar. Sections were post-stained for 5 min with lead citrate. Pictures were taken with a FEI Tecnai T12 transmission electron microscope at the Dunn School Bioimaging facility (University of Oxford). For analysis, photographs of randomly chosen entire grid squares were taken, the area of which totaled at least 20% of the total area of the cross section of the sciatic nerve. Macrophages were identified as cells containing foamy lipid bodies, and myelin ovoids as individual swirls of unraveling myelin. The numbers of macrophages, myelin ovoids, and myelinated and unmyelinated axons were counted from these montages of grid squares and normalized to square millimeters. Analysis was done using ImageJ Software. To calculate g-ratios, 10–20 electron micrographs were randomly picked using randomly generated numbers. All myelinated axons within each picture were measured to calculate the g-ratio, and all unmyelinated axons in each picture were measured for axon diameter using AxioVision LE Rel. 4.2 software. The examiner was blind to genotype.

## Behavioral analysis

The same designated room was used for all behavioral studies, and testing was performed at a consistent time of day. Mice were acclimatized to the testing equipment, and baseline values were obtained. All behavior studies were performed with the experimenter blind to treatment group or genotype. Baseline values were performed in order to determine that behavioral outcomes at baseline were within the normal range and no mice required exclusion from the behavioral analysis on this basis.

For SFI, mice were trained to run along a wooden 1-m-long catwalk into a dark box at the far end. The catwalk was covered with a piece of paper with the same dimensions, and the hind paws of the mice were inked with stamp ink. Mice ran along the catwalk, leaving their paw prints on the paper. The paper strips were left to dry, and measurements were manually taken. Paw prints used for measuring were chosen based on clarity and consistency and a succession of three or four prints at a point at which the mouse was walking at a moderate pace. Both uninjured and injured paw prints were measured. Measurements were as follows: toe spread, the distance between the first and fifth toes; intermediate toe spread, the distance between the second and fourth toes; and print length, the distance between the end of the third toe and the bottom of the hind pad. These measurements were used to calculate the SFI (Bain et al., 1989; Monte-Raso et al., 2008).

For the pin prick test, the skin of the foot was pinched by an insect pin (Austerliz) in the pads and toes in the lateral and medial parst. Sciatic nerve innervation of the skin in the lateral part (innervated by sural and tibial nerve) was measured as a reinnervation of the skin after sciatic nerve injury. Each pad or digit pinched was scored 0 for nonreflex withdrawal response of the paw, 1 for mild reflex withdrawal response, and 2 for good reflex withdrawal response.

## Statistics

All statistical analysis has been performed using GraphPad Prism 6 software. Data were reported as mean ± SEM, and the number of replicates was indicated in each figure legend. The level of significance was set at $P < 0.05$. The two-tailed unpaired Student's $t$ test was used to analyze differences between two groups. One-way ANOVA followed by the Tukey post hoc test was used for the analysis of more than two groups. To assess statistical differences in behavior experiments with multiple time points, repeated measures two-way ANOVA was performed with Sidak's multiple comparison tests. Distribution plots have been performed with Excel software, and the Kolmogorov-Smirnov test in SPSS has been used for statistical analysis of the distributions. Statistical analysis for each panel is specified in the figure legend. For parametric tests, data distribution was assumed to be normal, but this was not formally tested. The experimenters were blind to group/treatment during data acquisition and analysis before unblinding for statistical analysis.

## Online supplemental material

Fig. S1 shows the phenotypic outcome of injured $RalB^{-/-}$ mice. Fig. S2 shows the phenotypic outcome of injured $cRalA$ mice. Fig. S3 shows differences in comparing the effects of active RalA versus RalB on SC process formation. Fig. S4 shows transduction efficiency of the lentivirus. Fig. S5 shows the effect of active RalB on SC migration and axon myelination.

# Acknowledgments

We thank Dr. Errin Johnson and Anna Pielach from the Electron Microscopy Facility (University of Oxford, Oxford, UK) for their helpful assistance with EM, Dr. Gregory Weir and Dr. Georgios Baskozos for their help in the review of the manuscript, and Dr. Pascal Peschard and Chris Marshall for providing $RalA^{fl/fl}$ and $RalB^{-/-}$ mice.

This work was supported by Wellcome Trust Clinical Scientist Grant awards 202747/Z/16/Z and 095698z/11/z.

The authors declare no competing financial interests.

Author contributions: J. Galino and D.L.H. Bennett designed research; J. Galino, I. Cervellini, N. Zhu, N, Stöberl, M. Hütte, F.R. Fricker, and G. Lee performed research; J. Galino and I. Cervellini analyzed data; L. McDermott contributed to analytic tools; G. Lalli revised the paper and provided reagents, analytical tools, and critical analysis; and J. Galino and D.L.H. Bennett wrote the first draft paper with input from all authors on the final draft.

Submitted: 28 November 2018

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
