## [Reviewer comments · The Journal of Cell Biology]

RalGTPases contribute to Schwann cell repair after nerve injury via regulation of process formation

Jorge Galino, Ilaria Cervellini, Ning Zhu, Nina Stöberl, Meike Hütte, Florence Fricker, Garrett Lee, Lucy McDermott, Giovanna Lalli, and David Bennett

Corresponding Author(s): David Bennett, University of Oxford and David Bennett, University of Oxford

Review Timeline:	Submission Date:	2018-11-28
	Editorial Decision:	2019-01-10
	Revision Received:	2019-04-10
	Editorial Decision:	2019-05-01
	Revision Received:	2019-05-08

Monitoring Editor: Marc Freeman

Scientific Editor: Tim Spencer

Transaction Report:

DOI: <https://doi.org/10.1083/jcb.201811002>

January 4, 2019

Re: JCB manuscript #201811002

Prof. David LH Bennett
University of Oxford

Dear Prof. Bennett,

Thank you for submitting your manuscript entitled "RalGTPases control Schwann cell repair after nerve injury through regulation of process formation". Your manuscript has been assessed by expert reviewers, whose comments are appended below. Although the reviewers express potential interest in this work, significant concerns unfortunately preclude publication of the current version of the manuscript in JCB.

You will see that although all three reviewers have voiced some enthusiasm for the premise, reviewers #1 and #3 feel that the small effect sizes that you find in your analyses bring the physiological relevance of this proposed role for RalA/B in remyelination after injury into question. We hope that you will be able to provide further corroborating evidence or at least provide convincing arguments for why you feel this pathway and your mechanistic advance is sufficiently relevant for remyelination in vivo to warrant publication in JCB. Please note that any revised manuscript will be sent to these same reviewers for further assessment so it is essential that this point be convincingly addressed. We will also hope that you will be able to satisfy each of the other reviewer concerns in full.

Please let us know if you are able to address the major issues outlined above and wish to submit a revised manuscript to JCB. Note that a substantial amount of additional experimental data likely would be needed to satisfactorily address the concerns of the reviewers. Our typical timeframe for revisions is three to four months; if submitted within this timeframe, novelty will not be reassessed. We would be open to resubmission at a later date; however, please note that priority and novelty would be reassessed.

If you choose to revise and resubmit your manuscript, please also attend to the following editorial points. Please direct any editorial questions to the journal office.

GENERAL GUIDELINES:

Text limits: Character count is < 40,000, not including spaces. Count includes title page, abstract, introduction, results, discussion, acknowledgments, and figure legends. Count does not include materials and methods, references, tables, or supplemental legends.

Figures: Your manuscript may have up to 10 main text figures. To avoid delays in production, figures must be prepared according to the policies outlined in our Instructions to Authors, under Data Presentation, <http://jcb.rupress.org/site/misc/ifora.xhtml>. All figures in accepted manuscripts will be screened prior to publication.

*****IMPORTANT:** It is JCB policy that if requested, original data images must be made available. Failure to provide original images upon request will result in unavoidable delays in publication.

Please ensure that you have access to all original microscopy and blot data images before submitting your revision.***

Supplemental information: There are strict limits on the allowable amount of supplemental data. Your manuscript may have up to 5 supplemental figures. Up to 10 supplemental videos or flash animations are allowed. A summary of all supplemental material should appear at the end of the Materials and methods section.

If you choose to resubmit, please include a cover letter addressing the reviewers' comments point by point. Please also highlight all changes in the text of the manuscript.

Regardless of how you choose to proceed, we hope that the comments below will prove constructive as your work progresses. We would be happy to discuss them further once you've had a chance to consider the points raised. You can contact the journal office with any questions, cellbio@rockefeller.edu or call (212) 327-8588.

Thank you for thinking of JCB as an appropriate place to publish your work.

Sincerely,

Marc Freeman, PhD
Monitoring Editor
JCB

Tim Spencer, PhD
Deputy Editor
Journal of Cell Biology
ORCID: 0000-0003-0716-9936

Reviewer #1 (Comments to the Authors (Required)):

In this paper, the author report a novel role for RalA/B GTPases in promoting regeneration after injury by regulating Schwann cell process extension. The exocyst complex is suggested as effector of Ral GTPases.

This paper contains a huge amount of work that unfortunately resulted in documenting a limited biological effect as a) single mutant KO are dispensable for repair; b) double mutants display a transient and minor phenotype; c) Schwann cell based experiments suggest the exocyst complex as effector, but either gain of function or rescue experiments are not revealing a major effect in the claimed regulated function, the promotion of process extension.

The other concern I have is that Ral A expression is downregulated after crush injury, but instead it appears as a promoter of repair rather than a negative regulator. Viceversa, Ral B expression is upregulated, and it has not a major role in this process (only a few data in Figure 9 and Suppl. 4)

More in detail:

Figure 1.

I suggest to remove this Figure, but rather to introduce a scheme of the experimental design for each in vivo Figure/data.

Figure 2.

Panel A. I suggest to use another marker to normalize protein loading (GAPDH?). Calnexin is an ER protein, whose expression might be regulated after injury (as it is in development)

Panel B, C: In wild-type nerves, RalA is downregulated, whereas RalB is upregulated at all time points after crush. Thus, I would expect a role for RalB in promoting regeneration rather than RalA, unless RalA inhibits regeneration. Authors did not comment on this, and experiments in vitro using isolated SC mainly focus on RalA (Figure 8 and Supplementary)

Panel D. I don't understand the rationale of looking at the expression of RalA only in Schwann cells. If the explanation provided "Given that SCs are a major cellular component of the peripheral nerve,...." is meant in relation to the injury model, axons are also relevant after 10-15 days post crush.

Immunohistochemistry on teased fibers or transverse sections should also be performed to evaluate RalA/B expression (SC versus axons)

What about RalB in SC in vitro?

Supplementary Figs 2 and 3

Panel A: after how many days of tamoxifen administration western blot analysis has been performed? Again, a schematic representation of the experimental design for each figure should help.

These are contralateral uninjured nerves. What happens in cKO injured nerves for RalA and B expression? Since at least the RalA Floxed model is a conditional KO this experiment could help to figure out the cell specificity of downregulation and upregulation observed in Figure 2.

Supplementary Figure 3.

RalA and B expression should be assessed after tamoxifen-mediated PlpCreERT2 recombination in uninjured nerves. The Plp promoter has a very low efficiency in intact nerves.

Figure 3

A western blot showing RalA and B protein expression in double mutant nerves should be performed in injured and uninjured nerves.

Panel D. Mean g-ratio values should be reported in the text or legend and n= number of fibers analyzed per condition in addition to n= number of animals should be indicated.

About the conclusion that "the absence of Ral A and B in SCs affects radial growth and axial elongation": reduced internodal length might be related to axial growth but less myelinated fibers and hypomyelination might be related to remyelination/differentiation rather than radial growth (radial process extension). Remak bundles are normal.

Figure 7.

As the phenotype is not striking even in in vitro, I suggest to reduce the number and complexity of evaluation and categorize observation into a) radial lamellipodia per cell; b) axial lamellipodia per cell; c) process length. Having these three categories only will help to understand differences and biological significance. Scale bar is missing in panel H. Process length measure is missing. The type of substrate is missing. Laminin? This is a crucial information.

Supplementary Figure 4.

Lv-A72L and LvB23V seem to have different transduction efficiency considering GFP panels. Low efficiency for RalB. Western blot analysis of overexpressed constructs might be reported. In this condition it is difficult to claim a difference between Ral A and B.

Also, as before, I suggest to reduce the scored categories up to three.

Figure 8.

As before, I suggest to reduce the scored categories.

Lv-A72L seems to induce a macrophage-like phenotype, differently from Lv-D49E and Lv-D49N which instead look similar.

Panel K. Number of myelin segments should be counted. The density in addition to the area can change.

In general, cells appear sub-confluent, which makes difficult to count lamellipodia and measure process length

Figure 9.

This is the most relevant information, as Supplementary Fig. 4 and Figure 8 represent gain of function experiments in isolated rat SC. To compare different constructs, a transduction efficiency should be documented, which means not only more or less cells being transduced but also copy number per cell. Again, I suggest to simplify scoring categories as before.

Reviewer #2 (Comments to the Authors (Required)):

This study by Galino and colleagues examines the role of the small GTPases RalA and RalB in regenerative Schwann cells in peripheral nerves following crush injury. RalA and RalB have been implicated in a range of cellular functions such as proliferation, vesicle targeting and receptor-mediated endocytosis. The proteins signal mainly downstream of AKT and Ras and target one or more of several known effectors. These include RBP1, PhospholipaseD1 and the exocyst components Exoc2 and Exoc8.

The authors demonstrate that RalA and RalB are required for the proper regeneration, target innervation and functional recovery of peripheral nerves following nerve crush injury. The lack of RalA/B in Schwann cells does not affect their proliferation or survival, nor does it affect macrophage recruitment or myelin clearance.

The authors then provide evidence that the observed defects are caused by a reduced ability of Schwann cells to produce or stabilise radial and axial processes. This inability of RalA/B^{-/-} Schwann cells to produce radial processes in culture could be rescued by constitutive active RalA and by constitutive active mutant RalA that had lost its ability to interact with RBP1 (D49N). However, a

constitutive RalA that could not interact with Exoc2/ Exoc8 (D49E) did not rescue radial process formation in RalA/B mutant Schwann cells. These data strongly support a role for RalA/B signalling in radial process formation/stabilisation in regenerative Schwann cells. Interestingly and in support of their thesis, they show that Schwann cells transduced with constitutive active RalA stimulate myelination in an in vitro myelinating culture system.

This is a well-controlled study and the data strongly support the main conclusions of this paper.

Minor points

-Western blot results should really have molecular weight indications.

-Please consider changing the colour of fluorescent images in Figure 3H and Figure 8. The dark blue on a black background has no contrast and it is impossible to judge what is going on here. There is absolute no need to use blue. White on a black background provides the right contrast.

-Please use the official names for the exocyst components Exoc2 and Exoc8 (throughout).

-Figure 8 and 9 would benefit from a clearer labelling. Why not substitute A72L for CARalA and D49E for CARalA-Exocyst and D49N for CARalA-RBP1?

Reviewer #3 (Comments to the Authors (Required)):

The paper by Galino et al. describes a role for RalGTPases in nerve repair. Using mice that are null for RalB and conditionally deleted for RalA in Schwann cells (PLP-CreErt2), the authors demonstrate that ablation of both genes causes a delay in nerve regeneration and a defect in Schwann cell elongation, muscle innervation and functional regeneration. The authors exclude an effect of RalGTPases in myelin degradation or Schwann cell proliferation, but provide evidence for a defect in Schwann cell process formation and elongation. Using lentiviruses that express WT or mutated forms of RalA, the authors show in vitro that the interaction between RalA and the exocyst may be relevant for Schwann cell elongation, process formation and myelination. Overall, the paper is well done and the results are convincing. Unfortunately, however, the effect of RalA/B ablation in nerve regeneration is extremely mild (i.e. Fig. 3) raising some questions about the significance of RalGTPase role overall. As a result, the wording to report the main findings in the text, including the title, are stated too strongly.

Specific points:

1) The defects in regeneration shown in Fig. 3 are extremely mild. A larger field with more fibers should be shown in 3A. At what age is PLPcreERT2 activated with tamoxifen? It is important to clarify if RalA/B double deletion is achieved after development.

2) The in vitro data are generally convincing, except for Figure 8I which does not seem to reflect the quantitation shown in the graph. In addition, the number of internodes must be counted to confirm a difference in the number of actual myelin segments. The "myelin area" that was used cannot account for example for differences in myelin thickness or for the presence of many short internodes.

3) In figure 9 it should be commented that all RalB activation and all the RalA mutants were able to rescue most aspects of cell morphology counted, namely axial lamellipodia and prolongation and radial prolongation.

NUFFIELD DEPARTMENT OF
CLINICAL NEUROSCIENCES

David Bennett,
Professor of neurology and neurobiology,
Honorary consultant neurologist,
West Wing, Level 6
John Radcliffe Hospital, Oxford, OX3 9DU
www.ndcn.ox.ac.uk

Tel: +44(0)1865 231512 Email: david.bennett@ndcn.ox.ac.uk Fax: 01865 234 830

09.04.2019

Dear Dr Freeman,

We would like to resubmit the article 'RalGTPases contribute to Schwann cell repair after nerve injury via regulation of process formation' by Galino et al. for consideration for publication in the Journal of Cell Biology. Thank you for both your own and the reviewer's comments. In light of which we have extensively revised the paper to include new experiments/data. We note that Ueli Suter has recently resubmitted his paper on the developmental role of RalGTPase in Schwann cell development.

In terms of the general comment that you raise regarding the 'biological significance of our findings and small effect size'. It is not surprising to us that there is extensive redundancy in the system between the two RalA isoforms and this has been well documented in the literature. In the absence of both RalA and B signalling however there is a clear slowing of motor recovery following nerve injury and even at 1 month there are definite morphological differences with a larger g-ratio and shorter inter-nodal length in the animals lacking Ral GTPase signalling. Perception of effect size can depend on how you present the data for instance if you express % change in the number of axons of 1 micron diameter that are unmyelinated at 12 days and 1 month it's circa 100% between the genotypes. In relation to the *in vitro* data studying process outgrowth in SCs reviewer 3 actually comments that these findings are convincing and I think that reviewer 1 may have lost sight of this in the large amount of detail that we presented (we acknowledge that the *in vitro* analysis was complex and on this reviewers advice we have simplified our presentation). To take the number of radial lamellipodia in SCs as an example these virtually double as a consequence of expression of constitutively active RalA and in the absence of RalGTPase signalling are reduced by 85%. These are again large effect sizes.

Essentially all the reviewers comment on the extensive nature and high quality of the work. To further emphasise we performed and reported these *in vivo* experiments according to the ARRIVE guidelines (a recommendation according to JCB guidelines) and experimental method includes randomisation and full experimental blinding with appropriate statistical analysis. We have confidence therefore that not only are our findings biologically relevant but they are in no way inflated and will be reproducible by independent research groups. Below is our response to each of the reviewer's comments and we hope that the manuscript is now deemed suitable for publication in the Journal of Cell Biology.

Yours sincerely,

Professor David Bennett

Reviewer #1 (Comments to the Authors (Required)):

In this paper, the author report a novel role for RalA/B GTPases in promoting regeneration after injury by regulating Schwann cell process extension. The exocyst complex is suggested as effector of Ral GTPases.

This paper contains a huge amount of work that unfortunately resulted in documenting a limited biological effect as a) single mutant KO are dispensable for repair; b) double mutants display a transient and minor phenotype; c) Schwann cell based experiments suggest the exocyst complex as effector, but either gain of function or rescue experiments are not revealing a major effect in the claimed regulated function, the promotion of process extension.

We would respectfully disagree that the biological effects that we demonstrate are minor. Taking each point in turn:

- a) We agree that the mutation of a single RalA or B GTPase demonstrates that each is individually dispensable for repair and such redundancy is consistent with the many other cellular functions of Ral GTPase and well documented in the literature (Peschard P. et al, 2012; Wersäll A. et al 2018).
- b) The impact on mouse motor behaviour is transient although we would not argue that it is minor as it represents a significant difference in sciatic functional index (13 points at its peak) a robust measure of motor function. Furthermore assessing anatomical changes *in vivo* at 1 month post injury: there remains a significant increase in g-ratio, significant reduction in intermodal length and doubling in the number of axons with a diameter greater than 1 micron which are unmyelinated (see new Fig 3 and table 1).
- c) We are confused by this comment as 'not revealing an effect on the main regulated function' because the main regulated function that we report has been ignored: namely radial lamellipodia formation a key step in myelination (Bunge et al., 1989; Nodari et al., 2007; Montani et al., 2014). We see clear and significant effects on process formation with large effect sizes. Over-expression of constitutively active RalA in rat Schwann cells almost **doubles** the number of radial lamellipodia in these cells (see new figure 7 panel D). Furthermore the reverse experiment to assess the number of radial lamellipodia in Schwann cells in the absence of RalA signalling shows a decrease of **85%** (see new Figure 8 panel D) and although this effect can be completely rescued by expression of constitutively active RalA this rescue effect is absent in the RalA mutant than can't couple to exocyst.

Peschard, P., A. McCarthy, V. Leblanc-Dominguez, M. Yeo, S. Guichard, G. Stamp, and C.J. Marshall. 2012. Genetic deletion of RALA and RALB small GTPases reveals redundant functions in development and tumorigenesis. *Curr. Biol.* 22:2063–2068. doi:10.1016/j.cub.2012.09.013.

Wersäll A¹, Williams CM², Brown E², Iannitti T², Williams N², Poole AW². Mouse Platelet Ral GTPases Control P-Selectin Surface Expression, Regulating Platelet-Leukocyte Interaction. *Arterioscler Thromb Vasc Biol.* 2018 Apr; 38(4):787-800.

Bunge, R.P., M.B. Bunge, and M. Bates. 1989. Movements of the Schwann cell nucleus implicate progression of the inner (axon-related) Schwann cell process during myelination. *J. Cell Biol.* 109:273–84. doi:10.1083/JCB.109.1.273.

Nodari, A., D. Zambroni, A. Quattrini, F.A. Court, A. D'Urso, A. Recchia, V.L.J. Tybulewicz, L. Wrabetz, and M.L. Feltri. 2007. $\beta 1$ integrin activates Rac1 in Schwann cells to generate radial

lamellae during axonal sorting and myelination. *J. Cell Biol.* 177:1063–1075. doi:10.1083/jcb.200610014.

Montani, L., T. Buerki-Thurnherr, J.P. de Faria, J. a Pereira, N.G. Dias, R. Fernandes, A.F. Gonçalves, A. Braun, Y. Benninger, R.T. Böttcher, M. Costell, K.-A. Nave, R.J.M. Franklin, D. Meijer, U. Suter, and J.B. Relvas. 2014. Profilin 1 is required for peripheral nervous system myelination. *Development.* 141:1553–61. doi:10.1242/dev.101840.

The other concern I have is that Ral A expression is downregulated after crush injury, but instead it appears as a promoter of repair rather than a negative regulator. Viceversa, Ral B expression is upregulated, and it has not a major role in this process (only a few data in Figure 9 and Suppl. 4).

This is an interesting point and we agree that there does not appear to be a simple correlation between the total amount of RalGTPase isoforms at protein level and their functional relevance in terms of repair. We felt that this needed more exploration and so assessed the relative levels of activated Ral isoforms (ie GTP bound) as this is a more biologically relevant measure. RalGTPases activation state is correlated with their function, in contrast, activation state is not correlated with expression. Actually the increase in activation of RalGTPases can occur without any apparent increase in Ral expression and vice versa (Peschard et al 2012). In the case of RalA, even although total RalA is reduced at early time points after injury its activation status if anything increases (although this did not reach statistical significance). In the case of RalB the activation status is significantly reduced at longer time points (day 28 post injury). This new data has been added to new Figure 1.

Peschard, P., A. McCarthy, V. Leblanc-Dominguez, M. Yeo, S. Guichard, G. Stamp, and C.J. Marshall. 2012. Genetic deletion of RALA and RALB small GTPases reveals redundant functions in development and tumorigenesis. *Curr. Biol.* 22:2063–2068. doi:10.1016/j.cub.2012.09.013.

More in detail:

Figure 1.

I suggest to remove this Figure, but rather to introduce a scheme of the experimental design for each in vivo Figure/data. Thank you we believe that this recommendation will help the reader to follow the experiments in each figure of the manuscript so we have added a schema of the experiments in each figure and removed the old Figure 1 from the whole manuscript.

Figure 2.

Panel A. I suggest to use another marker to normalize protein loading (GAPDH?). Calnexin is an ER protein, whose expression might be regulated after injury (as it is in development). We agree that it's important to consider the stability of normalisers and we had chosen Calnexin because this protein is stable and not regulated by injury. In fact, its expression was not regulated in sciatic nerve 14 days after crush comparing injury vs sham (Ma Ki H. et al, 2018). We also report below the graph from our transcriptome analysis generated with data from <https://www.ncbi.nlm.nih.gov/geo/query/acc.cgi?acc=GSE106969> (Ma Ki H. et al, 2018). In addition, data in DRG neurons (which project through the sciatic nerve, data published in Baskozos et al.,2019) didn't show any difference in Calnexin, injury vs sham, 21 days after injury. More important, a proteomics database revealed no differences in Calnexin expression

after nerve injury (Barry AM et al, 2018). Finally, Calnexin has been used extensively by other groups as a normalizing protein during myelin development and after nerve injury (Triolo D et al, 2012; Ruirui Lu et al, 2011; Rivellini C. et al, 2012).

Ma Ki H., Duong P., Moran John J., Junaidi N. Svaren J. Polycomb repression regulates Schwann cell proliferation and axon regeneration after nerve injury. *Glia*. 2018. Nov: 66(11):2487-2502.

Baskozos G, Dawes JM, Austin JS, Antunes-Martins A, McDermott L, Clark AJ, Trendafilova T, Lees JG, McMahon SB, Mogil JS, Orengo C, Bennett DL. Comprehensive analysis of long noncoding RNA expression in dorsal root ganglion reveals cell-type specificity and dysregulation after nerve injury. *Pain*. 2019 Feb;160(2):463-485.

Barry Allison M., Sondermann Julia R., Sondermann Jan-Hendrik, Gómex-Varela David, Schmidt Manuela. Region-Resolved Quantitative Proteome Profiling Reveals Molecular Dynamics Associated With Chronic Pain in the PNS and Spinal Cord. *Front. Mol. Neurosci*. 2018 Aug; 14:11-259.

Daniela Triolo, Giorgia Dina, Carla Taveggia, Ilaria Vaccari, Emanuela Porrello, Cristina Rivellini, Teuta Domi, Rosa La Marca, Federica Cerri, Alessandra Bolino, Angelo Quattrini, Stefano Carlo Previtali. Vimentin regulates peripheral nerve myelination. *Development*. 2012 139: 1359-1367.

Ruirui Lu, Wiebke Kallenborn-Gerhardt, Gerd Geisslinger, Achim Schmidtko. Additive Antinociceptive Effects of a Combination of Vitamin C and Vitamin E after Peripheral Nerve Injury. *PLoS One*. 2011. 6(12):e29240.

Cristina Rivellini, Giorgia Dina, Emanuela Porrello, Federica Cerri, Marina Scarlato, Teuta Domi, Daniela Ungaro, Ubaldo Del Carro, Alessandra Bolino, Angelo Quattrini, Giancarlo Comi, Stefano C. Previtali. Urokinase Plasminogen Receptor and the Fibrinolytic Complex Play a Role in Nerve Repair after Nerve Crush in Mice, and in Human Neuropathies. *PLoS One*. 2012. 7(2):e32059.

Panel B, C: In wild-type nerves, RalA is downregulated, whereas RalB is upregulated at all time points after crush. Thus, I would expect a role for RalB in promoting regeneration rather than RalA, unless RalA inhibits regeneration. Authors did not comment on this, and experiments *in vitro* using isolated SC mainly focus on RalA (Figure 8 and Supplementary).

Regarding the dysregulated expression of Ral A and B *in vivo* following injury please see our comment above in which we assessed the GTP bound state of these proteins. Our *in vitro* experiments using Schwann cells were performed examining the effects of both RalA and RalB as reported in the new Fig 8, Suppl. 3, 4 and 5. In the migration experiment (new Fig. 7 F-G and Suppl 5A) and in the myelination experiment (new Fig. 7H-J and Suppl. 5B-D). We noted that RalA has the greater impact on processes of SC migration and myelination versus RalB. In terms of process outgrowth both could enhance radial lamellipodia number however we did note that RalA and not by RalB could increase radial prolongation number (Suppl. Fig. 3I).

Panel D. I dont understand the rationale of looking at the expression of RalA only in Schwann cells. If the explanation provided "Given that SCs are a major cellular component of the peripheral nerve,..." is meant in relation to the injury model, axons are also relevant after 10-15 days post crush. The purpose of panel D in old Figure 2 was to show the localisation of RalA in SCs. Our predominant interest was in the extensive remodelling of the Schwann cell membrane after nerve injury given the importance of Ral GTPases in cellular process extension. We found that RalA was present in the edge of the lamellipodia of SCs consistent with this hypothesis. This is confirmed by our new data generated using teased fibres (new Fig. 2) which suggest a predominant localisation of RalA in Schwann cells versus axons.

When we looked at the localization of RalA in primary cultured DRG neurons we saw that it remains localised to the nucleus and the cell body but with minimal transport to the axons/terminals hence our focus on Schwann cells.

Immunohistochemistry on teased fibers or transverse sections should also be performed to evaluate RalA/B expression (SC versus axons). We have followed this suggestion as far as possible and achieved immunostaining of RalA (in which this issue is most relevant given that we are using a conditional gene ablation strategy specifically in SCs). Unfortunately all the RalB antibodies available to us are suitable for Western blot analysis but perform poorly in immunostaining *in vivo*. This issue is however less relevant in the RalB mouse line as this is a constitutive gene ablation throughout the body and there is data validating this at protein level (both published (Peschard P. et al, 2012) and our own) and so cellular localisation is less of an issue.

The new Figure 2, Panel C now contains immunohistochemistry on teased fibers of WT; and PLPCre+RalA^{-/-}RalB^{-/-} (cRalA/B) stained with S100 and RalA antibodies. The figure shows that RalA is predominately expressed on SCs in the teased fibers and demonstrate that RalA is effectively ablated in SCs after tamoxifen treatment.

Non-injured nerve

Teased fibers of Ctrl (WT mice) and *cRaiA/B*^{-/-} from non-injured nerves. RalA is ablated in *cRaiA/B*^{-/-} mice and it is mainly expressed in the Schwann cells.

Teased fibers of Ctrl (WT mice) and *cRala/B^{-/-}* from injured nerves. RalA is ablated in *cRala/B^{-/-}* mice and it is mainly expressed in the Schwann cells.

Peschard, P., A. McCarthy, V. Leblanc-Dominguez, M. Yeo, S. Guichard, G. Stamp, and C.J. Marshall. 2012. Genetic deletion of RALA and RALB small GTPases reveals redundant functions in development and tumorigenesis. *Curr. Biol.* 22:2063–2068. doi:10.1016/j.cub.2012.09.013.

What about RalB in SC in vitro? E? Commercial RalB antibodies did not perform well in either *in vivo* or *in vitro* IHC experiments. In order to show the RalB expression in rat SCs we transduced them with a lentivirus containing a construct of RalB with a MYC epitope that enabled us to assess RalB localization in SCs. We observed that RalB is more cytoplasmic rather than nuclear and we did not see the expression in lamelliopodia that we had seen for RalA. We decided not to add this picture in the paper for space reasons.

Rat SCs transduced with CARaB virus. Staining for GFP, Myc and S100 was performed. RaLB, indicated by the Myc epitope localization, shows a cytoplasmic localization.

Supplementary Figs 2 and 3

Panel A: after how many days of tamoxifen administration western blot analysis has been performed? Again, a schematic representation of the experimental design for each figure should help. We performed the analysis of RaLGPases expression in naïve and injured tissues two months after tamoxifen induction (i.e one month after injury on injured mice). A schematic representation of the experimental design has been added to all figures related to *in vivo* experiments.

These are contralateral uninjured nerves. What happens in cKO injured nerves for RaA and B expression? Since at least the RaA Floxed model is a conditional KO this experiment could help to figure out the cell specificity of downregulation and upregulation observed in Figure 2. New Suppl. Figure 2 panel A shows no differences in RaLB expression in the whole sciatic nerve of cRaA mice in both injured and non-injured nerves.

Supplementary Figure 3.

RaA and B expression should be assessed after tamoxifen-mediated PlpCreERT2 recombination in uninjured nerves. The Plp promoter has a very low efficiency in intact nerves.

We performed a WB to detect RaA expression in non-injured and injured nerves of cRaA/B^{-/-} 2 months after the tamoxifen treatment (new Figure 2 panel B). There was an average decrease in RaA expression of circa 50% at this time point. Our experiments on teased fibers showed a clear decrease in SCs RaA staining after tamoxifen treatment (new Figure 2 panel F). This result suggests that the residual RaA observed in the WB of the nerve is due to the expression in other cell types such as fibroblasts and cells in the epineurium.

PLPCreERT² has been described in the literature as a good transgenic animal to drive mutagenesis on Schwann cells.(Doerflinger et al., 2003). Unfortunately they didn't make a proper quantification of the recombination. We performed immunohistochemistry analysis of transversal sections of sciatic nerve from PLPCreERT²; tdtomato mice which had been treated

using our protocol for Tamoxifen administration. Tdtomato is a Cre reporter allele that has a loxP-flanked STOP cassette preventing transcription of a CAG promoter-driven red fluorescent protein variant (tdTomato). When crossed with PLPCre tamoxifen treated mice, STOP cassette is removed and tdTomato is highly expressed on Schwann cells (see below). In our hands 89% of S100 profiles co-stained for tdTomato and there was no co-staining for neurofilament. These results confirm the high efficiency and the specificity of this promoter under our tamoxifen treatment basis.

Immunohistochemistry on sciatic nerve transversal sections of non injured PLPCre; tdtomato mice co-stained for NF200 (axon marker) or S100 (Schwann cell marker) (green) with tomato (red). Note the high level of colocalization between S100 and Tom when compared with the absence of colocalization between NF200 and tom.

Old suppl. Figure 3 has been deleted and results have been added to the new table 2 that summarises the EM results.

Doerflinger, N.H., W.B. Macklin, and B. Popko. 2003. Inducible site-specific recombination in myelinating cells. *Genesis*. 35:63–72. doi:10.1002/gene.10154.

Figure 3

A western blot showing RalA and B protein expression in double mutant nerves should be performed in injured and uninjured nerves. New Figure 2 panel B shows RalA expression on injured and non-injured nerves after tamoxifen induction in the cRalA/B^{-/-} mice. The ablation of RalB in the double mutant has not been repeated as the Panel B of Suppl. Figure 1 clearly shows already the ablation of RalB in RalB^{-/-} mice which is a constitutive KO model.

Panel D. Mean g-ratio values should be reported in the text or legend and n= number of fibers analyzed per condition in addition to n= number of animals should be indicated.

About the conclusion that "the absence of Ral A and B in SCs affects radial growth and axial elongation": reduced internodal length might be related to axial growth but less myelinated fibers and hypomyelination might be related to remyelination/differentiation rather than radial growth (radial process extension). Remak bundles are normal.

We added a new summary table of EM results where the g-ratio and the other measures assessed on EM experiments can be shown. Furthermore, we added in the table legend the number of fibers per animal that we have analysed plus the total number of mice per group. We appreciate that comments on radial and axial growth are best left to the later studies *in vitro* where we have direct effects on process outgrowth. We have therefore removed this sentence from this part of the manuscript.

Figure 7.

As the phenotype is not striking even *in vitro*, I suggest to reduce the number and complexity of evaluation and categorize observation into a) radial lamellipodia per cell; b) axial lamellipodia per cell; c) process length. Having these three categories only will help to understand differences and biological significance. Scale bar is missing in panel H. Process length measure is missing

The type of substrate is missing. Laminin? This is a crucial information.

We respectfully disagree (please see earlier comment) as this is a striking phenotype *in vitro*. In fact, as clearly showed in the new Fig. 6 D-E the number of axial and radial lamellipodia was reduced of almost 50% in cRAlA/B^{-/-} compared to control mice. This significant decrease, together with the significant decrease in axial prolongation length, showed in the Fig. 6G, is likely responsible for the decrease in remyelination, muscle reinnervation and motor function observed *in vivo*. Therefore, we strongly believe that a loss of function of these GTPases is phenotypically relevant. We have followed the reviewer's advice to simplify the analysis by decreasing the categories reported in the figure. A scale bar has been added to panel H together with the process length measure that was missed.

The type of substrate on primary SC cultures for both rats and mice was PDL plus Laminin as described in material and methods section.

Supplementary Figure 4.

Lv-A72L and LvB23V seem to have different transduction efficiency considering GFP panels. Low efficiency for RalB. Western blot analysis of overexpressed constructs might be reported. In this condition it is difficult to claim a difference between Ral A and B. Also, as before, I suggest to reduce the scored categories up to three.

We agree with the reviewer that in this figure was not representative and the GFP levels on the picture showed variation suggesting that the efficiency of transduction could be different depending on the constructs used.

We have now added more representative pictures in the Figure that is now Suppl. Fig.3.

We have taken advantage of the fact that both CARAlA and CARAlB have an integrated Myc epitope to undertake a more quantitative analysis: we performed a WB of the homogenate of rat SCs transduced with both viruses respectively and found that the level of Myc expression was the same in both conditions. This WB has been added to the new Suppl figure 3 panel A.

We also reduced the number of scored categories as suggested by the reviewer.

Figure 8.

As before, I suggest to reduce the scored categories.

Lv-A72L seems to induce a macrophage-like phenotype, differently from Lv-D49E and Lv-D49N which instead look similar.

We have chosen photomicrographs of constitutive active RalA transduction that are more representative. This new picture replaced the old one and is now shown on new Fig 7 panel A.

Panel K. Number of myelin segments should be counted. The density in addition to the area can change. We used this method to analyse myelination in co-cultures as it was already reported in our previous paper from Clark AJ. et al “Co-cultures with stem cell-derived human sensory neurons reveal regulators of peripheral myelination”. In this paper we extensively validated the method of analysis of myelin area per coverslip by comparing it with the calculation of the number of myelin segments in the same coverslips. We found a strong correlation of $r=0.926$ between the two methods showing they are highly comparable for the analysis of myelin quantification.

Alex J. Clark,¹ Malte S. Kaller,¹ Jorge Galino,¹ Hugh J. Willison,² Simon Rinaldi,¹ and David L. H. Bennett. Co-cultures with stem cell-derived human sensory neurons reveal regulators of peripheral myelination. *Brain*. 2017 Apr; 140(4): 898–913.

Thank you for this comment we have added a reference to this paper in the methods section.

In general, cells appear sub-confluent, which makes difficult to count lamellipodia and measure process length. We believe that the reviewer is referring to the old Fig.8A. We have now changed the pictures (as also suggested by reviewer 2) in order to provide a better contrast and to show that the count of lamellipodia and the measure of process length could be effectively performed at this cell confluence (New Fig. 7)

Figure 9.

This is the most relevant information, as Supplementary Fig. 4 and Figure 8 represent gain of

function experiments in isolated rat SC. To compare different constructs, a transduction efficiency should be documented, which means not only more or less cells being transduced but also copy number per cell. Again, I suggest to simplify scoring categories as before. We have measured the transduction efficiency in both mice and rat Schwann cells using two different methods. First we reported the percentage of S100 cells which are GFP positive. Then we measured the fluorescence intensity of the GFP signal in the single cells. As reported in the method section, the percentage of S100/GFP positive cells was calculated analysing 4 images for each condition, taken at 10X magnification for mouse SCs and 20X magnification for rat SCs. The intensity of GFP fluorescence in single cells was calculated analysing 50 cells for each condition and measured using the ImageJ software in pictures taken with exactly the same setting conditions.

The intensity, measured in pictures taken with the exact same settings, were reported as fold change versus the GFP intensity in the control (cell transduced with GFP lentivirus; in percentage). No difference in the transduction efficiency was found between the different lentiviruses. These results relating to transduction in rats and mice are now shown in Suppl. Fig.4.

Reviewer #2 (Comments to the Authors (Required)):

This study by Galino and colleagues examines the role of the small GTPases RalA and RalB in regenerative Schwann cells in peripheral nerves following crush injury. RalA and RalB have been implicated in a range of cellular functions such as proliferation, vesicle targeting and receptor-mediated endocytosis. The proteins signal mainly downstream of AKT and Ras and target one or more of several known effectors. These include RBP1, PhospholipaseD1 and the exocyst components Exoc2 and Exoc8.

The authors demonstrate that RalA and RalB are required for the proper regeneration, target innervation and functional recovery of peripheral nerves following nerve crush injury. The lack of RalA/B in Schwann cells does not affect their proliferation or survival, nor does it affect macrophage recruitment or myelin clearance.

The authors then provide evidence that the observed defects are caused by a reduced ability of Schwann cells to produce or stabilise radial and axial processes. This inability of RalA/B^{-/-} Schwann cells to produce radial processes in culture could be rescued by constitutive active RalA and by constitutive active mutant RalA that had lost its ability to interact with RBP1 (D49N). However, a constitutive RalA that could not interact with Exoc2/ Exoc8 (D49E) did not rescue radial process formation in RalA/B mutant Schwann cells. These data strongly support a role for RalA/B signalling in radial process formation/stabilisation in regenerative Schwann cells. Interestingly and in support of their thesis, they show that Schwann cells transduced with constitutive active RalA stimulate myelination in an in vitro myelinating culture system. This is a well-controlled study and the data strongly support the main conclusions of this paper.

Minor points

-Western blot results should really have molecular weight indications.

Thank you we have now added molecular weights on the immunoblots.

-Please consider changing the colour of fluorescent images in Figure 3H and Figure 8. The dark blue on a black background has no contrast and it is impossible to judge what is going on here. There is absolute no need to use blue. White on a black background provides the right contrast.

We have changed from blue to white in order to have more contrast on the new figures 3D and 7A.

-Please use the official names for the exocyst components Exoc2 and Exoc8 (throughout). We have now changed the name of components of the exocyst to the official nomenclature.

-Figure 8 and 9 would benefit from a clearer labelling. Why not substitute A72L for CARaA and D49E for CARaA-Exocyst and D49N for CARaA-RBP1? We have followed reviewer indications and changed all of them to have a clearer identification. Lv-A72L is now CARaA; Lv-RalB is now CARaB, Lv-D49E is now CARaA-EC and Lv-D49N is now CARaA-BP1.

Reviewer #3 (Comments to the Authors (Required)):

The paper by Galino et al. describes a role for RalGTPases in nerve repair. Using mice that are null for RalB and conditionally deleted for RalA in Schwann cells (PLP-CreErt2), the authors demonstrate that ablation of both genes causes a delay in nerve regeneration and a defect in Schwann cell elongation, muscle innervation and functional regeneration. The authors exclude an effect of RalGTPases in myelin degradation or Schwann cell proliferation, but provide evidence for a defect in Schwann cell process formation and elongation. Using lentiviruses that express WT or mutated forms of RalA, the authors show *in vitro* that the interaction between RalA and the exocyst may be relevant for Schwann cell elongation, process formation and myelination. Overall, the paper is well done and the results are convincing. Unfortunately, however, the effect of RalA/B ablation in nerve regeneration is extremely mild (i.e. Fig. 3) raising some questions about the significance of RalGTPase role overall. As a result, the wording to report the main findings in the text, including the title, are stated too strongly.

We think that our findings are biologically meaningful since there is a clear effect on the rate of motor recovery and even at 1 month post-injury there are morphological differences following ablation of RalGTPase signalling. Furthermore there is a striking phenotype when examining Schwann cells *in vitro* in relation to process extension and migration. We have changed the title to make clearer that this pathway is contributory rather than a single regulator so as to avoid any confusion ie: **'RalGTPases contribute to Schwann cell repair after nerve injury via regulation of process formation'**

Specific points:

1) The defects in regeneration shown in Fig. 3 are extremely mild. A larger field with more fibers should be shown in 3A.

New composition pictures to show a larger field of each genotype and age have been added to new figure 3A.

At what age is PLPcreERT2 activated with tamoxifen? It is important to clarify if RalA/B double deletion is achieved after development.

We performed the analysis of GTPases expression on naïve and injured tissue two months after tamoxifen induction (i.e one month after injury on injured mice). Tamoxifen induction was undertaken in adults (at the age of 8 weeks). New WBs and teased fiber experiments (in New figure 2B) show good ablation of RalB in the whole nerve of RalB^{-/-} mice and RalA ablation specifically in SCs of RalA^{-/-} and cRalA/B^{-/-} animals.

2) The in vitro data are generally convincing, except for Figure 8I which does not seem to reflect the quantitation shown in the graph. We have changed the intensity of the pictures and now we believe that the change shown after transduction with the activated forms of RalGTPases on SC migration is more convincing (New Fig. 7A).

In addition, the number of internodes must be counted to confirm a difference in the number of actual myelin segments. The "myelin area" that was used cannot account for example for differences in myelin thickness or for the presence of many short internodes.

As previously explained in our response to Reviewer 1, we used this method of myelin area measurement instead of the count of the number of myelin segment because we previously carefully compared these two assessments in a previous paper from Clark AJ. et al "Co-cultures with stem cell-derived human sensory neurons reveal regulators of peripheral myelination". There was a correlation of 0.926 between the two measures and so we can state that the two methods are very highly comparable. We apologise that we did not make this clear in the first version of the manuscript and this paper is now referenced in the methods.

We agree with the reviewer that this methods does not account for differences in the myelin thickness but to properly address differences in the myelin thickness this could not be performed with immunostaining but electron microscopy.

3) In figure 9 it should be commented that all RalB activation and all the RalA mutants were able to rescue most aspects of cell morphology counted, namely axial lamellipodia and prolongation and radial prolongation. Thank you for this comment we have emphasized the rescue effect of all the activated forms of Ral in the result section pertaining to new Fig. 8 (which was Fig. 9).

May 1, 2019

RE: JCB Manuscript #201811002R

Prof. David LH Bennett
University of Oxford
JR hospital
Oxford OX39DU
United Kingdom

Dear Prof. Bennett:

Thank you for submitting your revised manuscript entitled "RalGTPases contribute to Schwann cell repair after nerve injury via regulation of process formation". The paper has now been assessed by the original reviewers #1 and #3. As you will see, although reviewer #1 is satisfied by the revisions, reviewer #3 remains unconvinced by the effect size in the mutants and continues to contend that counting internodes is a more informative method for assessing remyelination. Regarding effect size, while we see the reviewer's point on this issue, we also acknowledge that the in vitro assays are convincing and, particularly for in vivo assays, it is difficult to say what represents a 'biologically meaningful' effect size so we feel that the current data are sufficient. In addition, while we agree with the reviewer that counting internodes will do a better job of accounting for shorter or thicker internodes, we do not feel that application of this methodology will change the underlying conclusions of the study.

Therefore, we would be happy to publish your paper in JCB pending final revisions necessary to meet our formatting guidelines (see details below).

A. MANUSCRIPT ORGANIZATION AND FORMATTING:

Full guidelines are available on our Instructions for Authors page, <http://jcb.rupress.org/submission-guidelines#revised>. **Submission of a paper that does not conform to JCB guidelines will delay the acceptance of your manuscript.**

1) Text limits: Character count for Articles and Tools is < 40,000, not including spaces. Count includes title page, abstract, introduction, results, discussion, acknowledgments, and figure legends. Count does not include materials and methods, references, tables, or supplemental legends.

2) Figures limits: Articles and Tools may have up to 10 main text figures.

3) Figure formatting: Scale bars must be present on all microscopy images, including inset magnifications. Molecular weight or nucleic acid size markers must be included on all gel electrophoresis.

4) Statistical analysis: Error bars on graphic representations of numerical data must be clearly described in the figure legend. The number of independent data points (n) represented in a graph

must be indicated in the legend. Statistical methods should be explained in full in the materials and methods. For figures presenting pooled data the statistical measure should be defined in the figure legends. Please also be sure to indicate the statistical tests used in each of your experiments (both in the figure legend itself and in a separate methods section) as well as the parameters of the test (for example, if you ran a t-test, please indicate if it was one- or two-sided, etc.). Also, since you used parametric tests in your study (e.g. t-tests, ANOVA, etc.), you should have first determined whether the data was normally distributed before selecting that test. In the stats section of the methods, please indicate how you tested for normality. If you did not test for normality, you must state something to the effect that "Data distribution was assumed to be normal but this was not formally tested."

5) Materials and methods: Should be comprehensive and not simply reference a previous publication for details on how an experiment was performed. Please provide full descriptions (at least in brief) in the text for readers who may not have access to referenced manuscripts.

6) Please be sure to provide the sequences for all of your primers/oligos and RNAi constructs in the materials and methods. You must also indicate in the methods the source, species, and catalog numbers (where appropriate) for all of your antibodies.

7) Microscope image acquisition: The following information must be provided about the acquisition and processing of images:

- a. Make and model of microscope
- b. Type, magnification, and numerical aperture of the objective lenses
- c. Temperature
- d. imaging medium
- e. Fluorochromes
- f. Camera make and model
- g. Acquisition software
- h. Any software used for image processing subsequent to data acquisition. Please include details and types of operations involved (e.g., type of deconvolution, 3D reconstitutions, surface or volume rendering, gamma adjustments, etc.).

8) References: There is no limit to the number of references cited in a manuscript. References should be cited parenthetically in the text by author and year of publication. Abbreviate the names of journals according to PubMed.

9) Supplemental materials: There are strict limits on the allowable amount of supplemental data. Articles/Tools may have up to 5 supplemental figures. Please also note that tables, like figures, should be provided as individual, editable files. A summary of all supplemental material should appear at the end of the Materials and methods section.

10) Conflict of interest statement: JCB requires inclusion of a statement in the acknowledgements regarding competing financial interests. If no competing financial interests exist, please include the following statement: "The authors declare no competing financial interests." If competing interests are declared, please follow your statement of these competing interests with the following statement: "The authors declare no further competing financial interests."

11) ORCID IDs: ORCID IDs are unique identifiers allowing researchers to create a record of their various scholarly contributions in a single place. At resubmission of your final files, please consider providing an ORCID ID for as many contributing authors as possible.

B. FINAL FILES:

-- High-resolution figure and video files: See our detailed guidelines for preparing your production-ready images, <http://jcb.rupress.org/fig-vid-guidelines>.

Thank you for this interesting contribution, we look forward to publishing your paper in Journal of Cell Biology.

Sincerely,

Marc Freeman, PhD
Monitoring Editor
JCB

Tim Spencer, PhD
Deputy Editor
Journal of Cell Biology

Reviewer #1 (Comments to the Authors (Required)):

The authors addressed all my concerns and I think that the results are now more convincing and

clearly presented. The role of the exocyst complex as effector of Ral-GTPases is well supported with a combination of experiments performed using rat isolated Schwann cells in a gain-of-function setting; Schwann cell/DRG neuron co-cultures, and mutant mouse isolated Schwann cells.

Reviewer #3 (Comments to the Authors (Required)):

The revised paper by Galino et al. does not provide significant data that further strengthen the original findings. The points remain that

1) the phenotype is mild.

2) the DRG myelination figure is not convincing, and the number of internodes should be counted.

Despite the correlation shown in the Clark paper, this method does not account for shorter internodes or thicker internodes. The biological data provided by this method only is incomplete.